# Identification of electroporation sites in the complex lipid organization of the plasma membrane

**Lea Rems[1,2]\*, Xinru Tang[1,3]†, Fangwei Zhao[1,3]†, Sergio Pérez-Conesa[1], Ilaria Testa[1], Lucie Delemotte[1]\***

[1]KTH Royal Institute of Technology, Dept. Applied Physics, Science for Life Laboratory, Solna, Sweden; [2]University of Ljubljana, Faculty of Electrical Engineering, Ljubljana, Slovenia; [3]University of Chinese Academy of Sciences, Beijing, China

**Abstract** The plasma membrane of a biological cell is a complex assembly of lipids and membrane proteins, which tightly regulate transmembrane transport. When a cell is exposed to strong electric field, the membrane integrity becomes transiently disrupted by formation of trans-membrane pores. This phenomenon termed electroporation is already utilized in many rapidly developing applications in medicine including gene therapy, cancer treatment, and treatment of cardiac arrhythmias. However, the molecular mechanisms of electroporation are not yet sufficiently well understood; in particular, it is unclear where exactly pores form in the complex organization of the plasma membrane. In this study, we combine coarse-grained molecular dynamics simulations, machine learning methods, and Bayesian survival analysis to identify how formation of pores depends on the local lipid organization. We show that pores do not form homogeneously across the membrane, but colocalize with domains that have specific features, the most important being high density of polyunsaturated lipids. We further show that knowing the lipid organization is sufficient to reliably predict poration sites with machine learning. Additionally, by analysing poration kinetics with Bayesian survival analysis we show that poration does not depend solely on local lipid arrangement, but also on membrane mechanical properties and the polarity of the electric field. Finally, we discuss how the combination of atomistic and coarse-grained molecular dynamics simulations, machine learning methods, and Bayesian survival analysis can guide the design of future experiments and help us to develop an accurate description of plasma membrane electroporation on the whole-cell level. Achieving this will allow us to shift the optimization of electroporation applications from blind trial-and-error approaches to mechanistic-driven design.

**\*For correspondence:**
lea.rems@fe.uni-lj.si (LR);
lucie.delemotte@scilifelab.se
(LD)

†These authors contributed
equally to this work

**Competing interest:** See page
20

**Reviewing Editor:** Qiang Cui,
Boston University, United States

## Introduction

The plasma membrane of a cell is a complex assembly of hundreds of different types of lipids and membrane proteins, which tightly regulate transmembrane trafficking and participate in cell signalling (*Krapf, 2018*; *van Meer et al., 2008*). The molecular organization of the plasma membrane and its integrity are essential for the life of the cell. However, when the cell is exposed to external forces, the membrane integrity can become transiently disrupted by formation of transmembrane pores. Such disruption can be useful in many clinical applications, for example when nucleic acids need to be delivered across the plasma membrane into the cell interior, where they can carry out their tasks (*Gary and Weiner, 2020*; *Glass et al., 2018*). Various physical methods can induce transmembrane pores, including ultrasound, light, electric field, and mechanical deformation (stretching/squeezing) (*Gurtovenko et al., 2010*; *Ding et al., 2017*; *Yang et al., 2020*; *Schneckenburger, 2019*). In terms of clinical applications, poration by the application of electric fields or *electroporation* is the most widely

used method. It is approved for treatment of solid tumours, and it is being tested in clinical trials for gene therapy, vaccination against cancer and infectious diseases, and for cardiac ablation (*Campana et al., 2019*; *Geboers et al., 2020*; *Algazi et al., 2020*; *McBride et al., 2021*).

One of the bottlenecks of electroporation is that its protocol, especially the parameters of the applied electric pulses (amplitude, duration, number, repetition rate), needs to be optimized for each specific application and also for each specific cell/tissue type (*Cemazar et al., 2009*; *Rols and Teissié, 1998*; *Hunter et al., 2021*). When using electroporation for intracellular delivery of nucleic acids, the cells need to survive the treatment to be able to express the transgene. On the contrary, when using electroporation as an ablation modality, the cells need to die. Any electroporation-based treatment thus needs to be designed to either avoid or reach the point of no return leading to cell death. At present, the understanding of this point of no return is very limited (*Batista Napotnik et al., 2021*). One of the main reasons for this is our insufficient understanding of the molecular mechanisms that govern the increased cell membrane permeability induced by the applied electric field (*Kotnik et al., 2019*). If we can identify the molecular alterations of the cell membrane, we can begin to connect them to the biological response of the cells to pulsed electric fields.

The most accepted models, that describe electroporation on the whole-cell level, consider that pores can form only in the lipid domains of the plasma membrane and that all pores exhibit a similar kinetic behaviour (*Krassowska and Filev, 2007*; *Li and Lin, 2011*; *Gowrishankar et al., 2013*). However, accumulating evidence from experiments and simulations on model systems speaks against these assumptions. Poration kinetics in pure lipid bilayers has been shown to depend on the type of lipids and their phase state (*Perrier et al., 2017*; *Sengel and Wallace, 2016*). Since the lipids in the plasma membrane organize in domains (*Lu and Fairn, 2018*; *Levental et al., 2020*), there must exist locations which are more and less prone to poration. Moreover, our research suggests that pores can nucleate within some membrane proteins, causing protein denaturation and lipid rearrangement (*Rems et al., 2020*). Such lipid/protein pores can be more stable than pure lipid pores and are more likely to explain the persistent increase in plasma membrane permeability following exposure to electric pulses. Studies have further shown that pore formation and/or expansion is affected by the actin cytoskeleton, either via actin's influence on lipid organization or the mechanical properties of the membrane (*Muralidharan et al., 2021*; *Perrier et al., 2019*). The current challenge is to gather this ensemble of findings into a coherent and predictive mathematical model describing electroporation of the living cell's plasma membrane. In a living cell's plasma membrane, pores cannot form anywhere: as soon as a sufficient number of pores are formed, the transmembrane voltage drops, preventing formation of new pores (*DeBruin and Krassowska, 1999*; *Smith et al., 2014*). In other words, pores will form preferentially in specific sites with the highest poration propensity. However, it remains to be elucidated which are the properties of these sites.

The challenge of studying pores in the plasma membrane experimentally is that pores are nanometre-sized and open only transiently, whereby most of them appear to rapidly close (ns–μs range) after turning off the electric field (*Melikov et al., 2001*; *Bennett et al., 2014*; *Sözer et al., 2020*). Pores have been imaged in giant unilamellar vesicles (*Riske and Dimova, 2005*; *Lira et al., 2021*); however, these pores have reached sizes on the order of 1 μm, which have not been observed in cell plasma membranes, likely because the actin cytoskeleton limits pore expansion (*Perrier et al., 2019*). Pores have also been imaged in real time in droplet interface bilayers with TIRF (total internal reflection fluorescence) microscopy (*Sengel and Wallace, 2016*); however, the membranes were exposed to seconds-long electric pulses, which are much longer than pulses used in electroporation applications (ns–ms range), and which would likely not be tolerated by living cells. A few attempts have been made to visualize pores in cells using electron microscopy (*Chang and Reese, 1990*; *Lee et al., 2012*); however, the observed pores were suggested to be artefacts of sample preparation (*Teissie et al., 2005*). Overall, the current state of experimental methods does not appear to be at a stage where it would provide the spatiotemporal resolution required to understand the molecular mechanisms of plasma membrane electroporation in its entirety.

In this study, we thus resort to molecular modelling methods to investigate plasma membrane electroporation. In particular, we use coarse-grained molecular dynamics simulations, building on their success in studying membrane lateral organization and dynamic behaviour (*Duncan et al., 2017*; *Marrink et al., 2019*; *Khalid and Rouse, 2020*). By running electroporation simulations on lipid membranes mimicking the realistic composition of plasma membranes, we confirm that pores do not

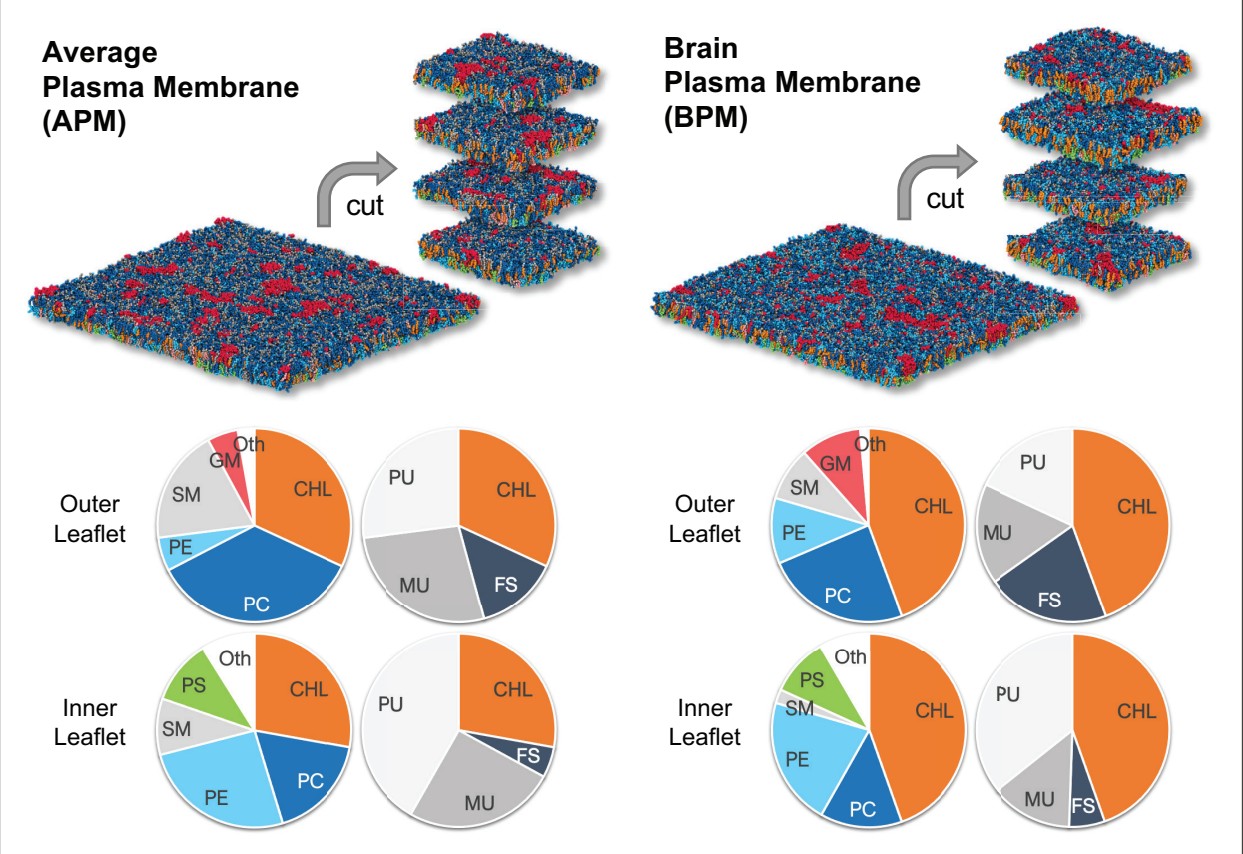

**Figure 1.** The average mammalian and human brain plasma membranes. Two equilibrated membranes taken from the study of *Ingólfsson et al., 2017* were cut into four 30 nm x 30 nm large pieces each. The pie charts show the fraction of lipid subgroups in the inner and outer leaflets of the membranes. CHL, cholesterol; PC, phosphatidylcholine; PE, phosphatidylethanolamine; SM, sphingomyelin; GM, gangliosides; PS, phosphatidylserine; FS, fully saturated lipids; MU, monounsaturated lipids; PU, polyunsaturated lipids. Figure inspired by *Ingólfsson et al., 2017*.

The online version of this article includes the following figure supplement(s) for figure 1:

**Figure supplement 1.** Fraction of different lipidsin the inner and outer membrane leaflets.

form homogeneously across the membrane, but colocalize with domains that have specific features, particularly high content of polyunsaturated (PU) lipids. By training machine learning algorithms, we further demonstrate that knowing the local lipid distribution is sufficient to predict with ~80–90% accuracy the locations, which are most likely to be porated. Additionally, by analysing poration kinetics with Bayesian survival analysis we show that poration does not depend solely on local lipid arrangement, but also on membrane mechanical properties and the polarity of the electric field. Finally, we discuss how atomistic and coarse-grained molecular dynamics simulations, machine learning methods, and Bayesian survival analysis combined can help us develop more accurate cell-level models, which are required to foster new and better electroporation-based applications.

## Results

To study plasma membrane electroporation we have used coarse-grained membranes consisting of >60 different lipid types parametrized with the Martini force field (*Marrink et al., 2007*; *de Jong et al., 2013*). The membranes mimic the composition of either an idealized average mammalian plasma membrane (APM) or a human brain plasma membrane (BPM) and have been developed and equilibrated in earlier work (*Ingólfsson et al., 2014*; *Ingólfsson et al., 2017*). The lipid composition of both APM and BPM is asymmetric, the lipids in the outer leaflet being different from those of the inner leaflet. Both compositions contain similar lipid types but differ in their fractions (*Figure 1*, *Figure 1—figure supplement 1*, *Tables 1 and 2*).

**Table 1.** Separation of lipids into groups based on their headgroup type or tail saturation.
The lipids are grouped into subtypes, depending on their headgroup type and tail saturation. Lipids are considered to be fully saturated, if they contain no double bonds in any of the tails. Lipids are considered to be monounsaturated, if they contain exactly one double bond in one or both of the tails. Lipids are considered to be polyunsaturated, if they contain at least two double bonds in one or both of the tails[*].

| Group name | Abbrev. | Martini lipids |
|---|---|---|
| Phosphatidylcholines | PC | DAPC, DOPC, DPPC, OIPC, OUPC, PAPC, PEPC, PFPC, PIPC, POPC, PUPC |
| Phosphatidylethanolamines | PE | DAPE, DOPE, DUPE, OAPE, OIPE, OUPE, PAPE, PIPE, POPE, PQPE, PUPE |
| Sphingomyelins | SM | BNSM, DBSM, DPSM, DXSM, PBSM, PGSM, PNSM, POSM, XNSM |
| Gangliosides | GM | DBG1, DPG1, DXG1, PNG1, POG1, XNG1, DBG3, DPG3, DXG3, PNG3, POG3, XNG3, DBGS, DPGS, PNGS, POGS |
| Ceramides | CE | DBCE, DPCE, DXCE, PNCE, POCE, XNCE |
| Lysolipids | LPC | APC, IPC, OPC, PPC, UPC, IPE, PPE |
| Diglycerides | DAG | PODG, PIDG, PADG, PUDG |
| Phosphatidylserines | PS | DAPS, DOPS, DPPS, DUPS, OUPS, PAPS, PIPS, POPS, PQPS, PUPS |
| Phosphatidylinositols | PI | POPI, PIPI, PAPI, PUPI |
| Phosphatic acids | PA | POPA, PIPA, PAPA, PUPA |
| Phosphatidylinositol phosphates | PIP | PAP1, PAP2, PAP3, POP1, POP2, POP3 |
| Cholesterol | CHOL | CHOL |
| Fully saturated tails | FS | DPPC, DBSM, DPSM, DXSM, PBSM, DPPS, DBCE, DPCE, DXCE, PPC, PPE, DBG1, DPG1, DXG1, DBG3, DPG3, DXG3, DBGS, DPGS |
| Monounsaturated tails | MU | DOPC, POPC, DOPE, POPE, BNSM, PGSM, PNSM, POSM, XNSM, DOPS, POPS, POPI, POP1, POP2, POP3, POPA, PODG, PNCE, POCE, XNCE, OPC, PNG1, POG1, XNG1, PNG3, POG3, XNG3, PNGS, POGS |
| Polyunsaturated tails | PU | OIPC, OUPC, PAPC, PEPC, PFPC, PIPC, PUPC, OAPE, OIPE, OUPE, PAPE, PIPE, PQPE, PUPE, OUPS, PAPS, PIPS, PQPS, PUPS, PAPI, PIPI, PUPI, PAP1, PAP2, PAP3, PAPA, PIPA, PUPA, PADG, PIDG, PUDG, APC, IPC, UPC, IPE, DAPC, DUPE, DAPE, DAPS, DUPS |

[*]The way in which lipids are grouped by tail saturation is different than in **Ingólfsson et al., 2017**, where the grouping is based on the total number of double bonds in both lipid tails. Here, the grouping is motivated by the role of lipid oxidation in electroporation, whereby lipid tails containing two or more double bonds are considerably more prone to oxidative damage than tails containing a single double bond. This is because bis-allylic hydrogens are much more easily abstracted by free radicals compared to allylic hydrogens (**Reis and Spickett, 2012**). Furthermore, membranes made of polyunsaturated lipids (by our definition) were found to be considerably more prone to poration/rupture by mechanical stretching compared to membranes made of lipids containing a single bond in one or both lipid tails (**Olbrich et al., 2000**). Thus, we consider that a lipid is polyunsaturated only if it contains at least one polyunsaturated tail.

The following subsections present the results and analysis as follows. First, we present the results from electroporation simulations and demonstrate that all membranes exhibit preferential poration sites. Then we determine local membrane properties and examine which of them increases/decreases the poration propensity. We further investigate the importance of local membrane properties by training machine learning algorithms and predicting the sites, which are most likely to be porated. Finally, we apply Bayesian survival analysis to investigate how membrane properties influence the poration kinetics and to develop an underlying kinetic model.

**Table 2.** The total number of lipids and the percentage of individual lipid groups in each of the eight membranes.

The numbers correspond to inner/outer leaflet.

| | Average plasma membrane | | | | Brain plasma membrane | | | |
|---|---|---|---|---|---|---|---|---|
| | mem #1 | mem #2 | mem #3 | mem #4 | mem #1 | mem #2 | mem #3 | mem #4 |
| Total number of lipids in each leaflet | | | | | | | | |
| in / out | 1503/1638 | 1564/1663 | 1571/1650 | 1553/1663 | 1728/1891 | 1740/1847 | 1770/1886 | 1764/1906 |
| Percentage of lipid groups | | | | | | | | |
| CHOL | 27.7/31.7 | 30.4/32.1 | 30.8/32.1 | 29.6/33.0 | 44.5/45.7 | 45.1/46.1 | 45.4/46.3 | 44.4/46.4 |
| PC | 17.9/36.4 | 18.9/34.3 | 15.9/34.7 | 17.8/34.3 | 12.4/23.0 | 12.4/23.6 | 14.8/25.8 | 14.2/21.9 |
| PE | 24.5/6.6 | 21.9/4.6 | 24.8/5.3 | 25.2/6.2 | 22.8/9.7 | 21.8/11.7 | 19.6/9.2 | 20.9/10.5 |
| SM | 10.6/18.7 | 8.9/19.5 | 9.7/19.6 | 8.6/19.1 | 2.1/8.9 | 2.1/9.0 | 2.3/9.2 | 2.8/8.5 |
| GM | 0.0/3.7 | 0.0/7.0 | 0.0/5.2 | 0.0/4.7 | 0.0/11.3 | 0.0/8.6 | 0.0/8.0 | 0.0/11.5 |
| PS | 11.3/0.0 | 10.7/0.0 | 9.6/0.0 | 10.6/0.0 | 9.8/0.0 | 11.0/0.0 | 9.1/0.0 | 8.7/0.0 |
| PI | 3.7/0.0 | 4.5/0.0 | 5.0/0.0 | 5.0/0.0 | 5.4/0.0 | 4.9/0.0 | 5.0/0.0 | 5.1/0.0 |
| PIP | 2.0/0.0 | 1.7/0.0 | 1.1/0.0 | 1.5/0.0 | 1.4/0.0 | 0.9/0.0 | 1.3/0.0 | 1.7/0.0 |
| PA | 1.6/0.0 | 2.1/0.0 | 2.0/0.0 | 0.9/0.0 | 0.3/0.0 | 0.6/0.0 | 0.5/0.0 | 0.2/0.0 |
| LPC | 0.0/1.5 | 0.0/0.8 | 0.0/1.4 | 0.0/1.0 | 0.5/0.3 | 0.2/0.5 | 0.5/0.6 | 0.3/0.3 |
| DAG | 0.5/1.0 | 0.3/0.5 | 0.4/0.8 | 0.3/1.0 | 0.4/0.2 | 0.3/0.1 | 0.8/0.3 | 1.0/0.1 |
| CE | 0.3/0.4 | 0.6/1.1 | 0.6/0.9 | 0.6/0.7 | 0.2/0.7 | 0.6/0.3 | 0.7/0.5 | 0.7/0.7 |
| FS | 6.6/12.2 | 4.4/14.8 | 5.3/14.7 | 4.8/14.0 | 5.1/22.4 | 5.2/19.3 | 6.1/19.8 | 5.8/20.6 |
| MU | 24.6/26.7 | 25.4/29.5 | 24.4/27.1 | 25.4/26.6 | 13.2/15.5 | 12.9/15.9 | 14.1/16.6 | 15.2/17.2 |
| PU | 41.1/29.4 | 39.8/23.6 | 39.5/26.2 | 40.2/26.5 | 37.2/16.3 | 36.8/18.7 | 34.4/17.2 | 34.5/15.8 |

## Membranes exhibit preferential poration sites

The original APM and BPM membranes from Ingolfsson et al. (*Ingólfsson et al., 2014*; *Ingólfsson et al., 2017*) were cut into four pieces each (*Figure 1*) to increase sampling, since the analysis was focused on the first poration event. After a short re-equilibration, each of the eight membranes was subject to electroporation simulations, 60 simulations under hyperpolarizing field and 60 simulations under depolarizing electric field. Both polarities of the electric field were considered, since during exposure of a cell to an electric field, the plasma membrane becomes hyperpolarized on the side facing the positive electrode and depolarized on the side facing the negative electrode. The electric field was set to ±127.7 mV/nm, which was high enough to induce a pore within ~15 ns. Being able to observe poration over short time scales was important to minimize lateral lipid diffusion and provide a reliable mapping of local membrane features before electric field application to the likelihood of a poration event. Note that the value of the electric field strength imposed in simulations is not directly comparable to the electric field strengths reported in experimental studies (see section Molecular dynamics simulations for further explanation). After identifying when and where the pores formed in each simulation, we observed that pore locations do not distribute homogeneously along the membrane surface but often cluster together (*Figure 2*). An individual membrane exhibits one or more of such clusters. The location of pores is similar albeit not completely identical under depolarizing and hyperpolarizing electric field.

## Poration sites colocalize with domains with specific features, in particular with a high density of PU lipids

We hypothesized that pores preferably form in nanodomains with specific features. We used the recently developed tool MemSurfer (*Bhatia et al., 2019*) to extract from each membrane the local

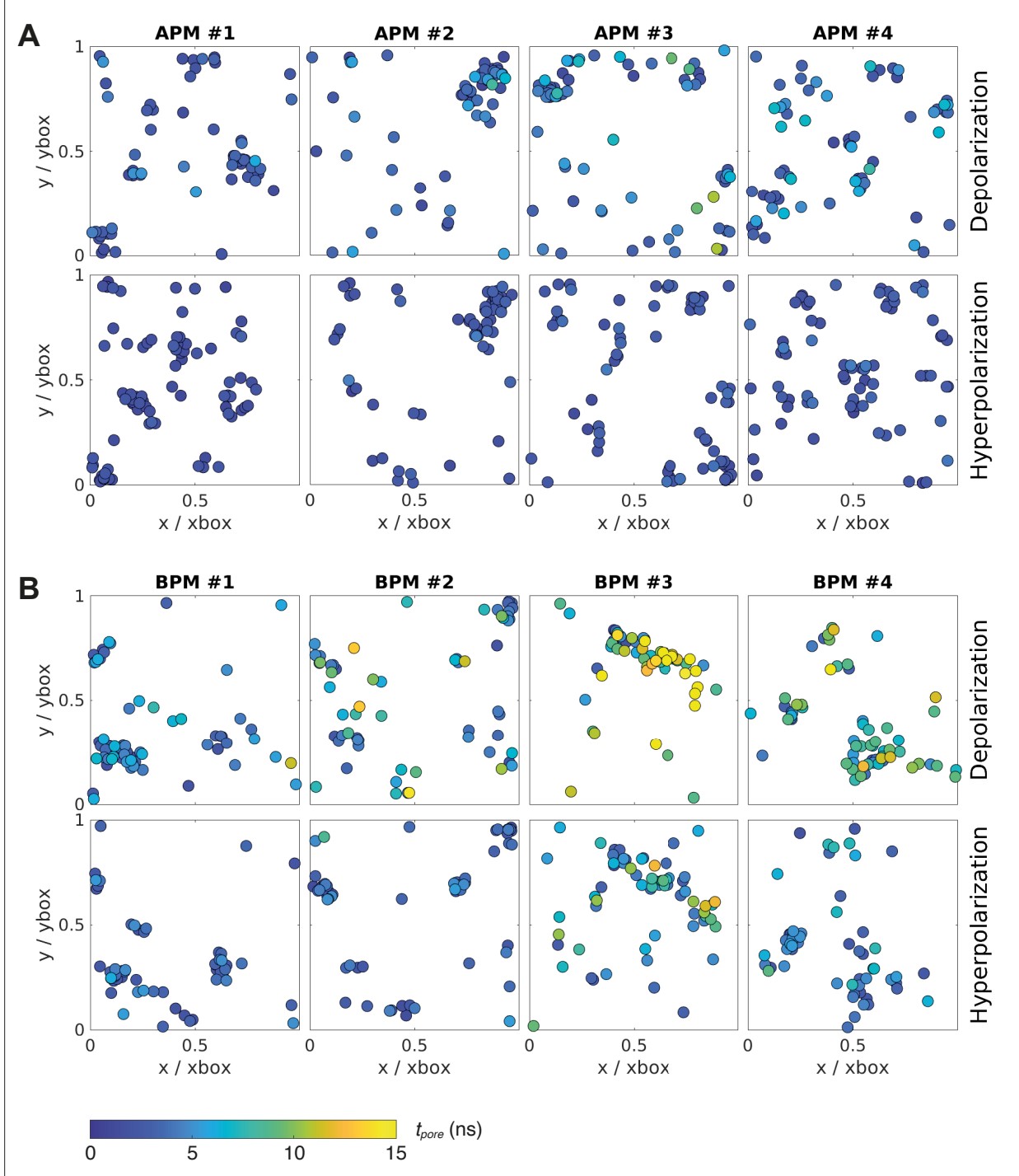

**Figure 2.** Location and kinetics of pore formation in each membrane upon depolarization and hyperpolarization. The pore locations are expressed relative to the dimensions of the simulation box at the poration time, to correct for the natural fluctuations in the membrane area during the simulation. All data points correspond to the first poration event. The colour of the circle codes for poration time according to the colour bar. (**A**) Average plasma membranes. (**B**) Brain plasma membranes.

area per lipid (APL), membrane thickness, mean curvature, cosine of the dipole angle (cos $\theta_{dip}$), charge, and lipid tail order parameter (*Figure 3—figure supplement 1*). We also determined the local density of individual groups of lipids, grouping the lipids either according to their head architecture or their tail saturation. These membrane features were extracted from 101 frames of a 10-ns long trajectory

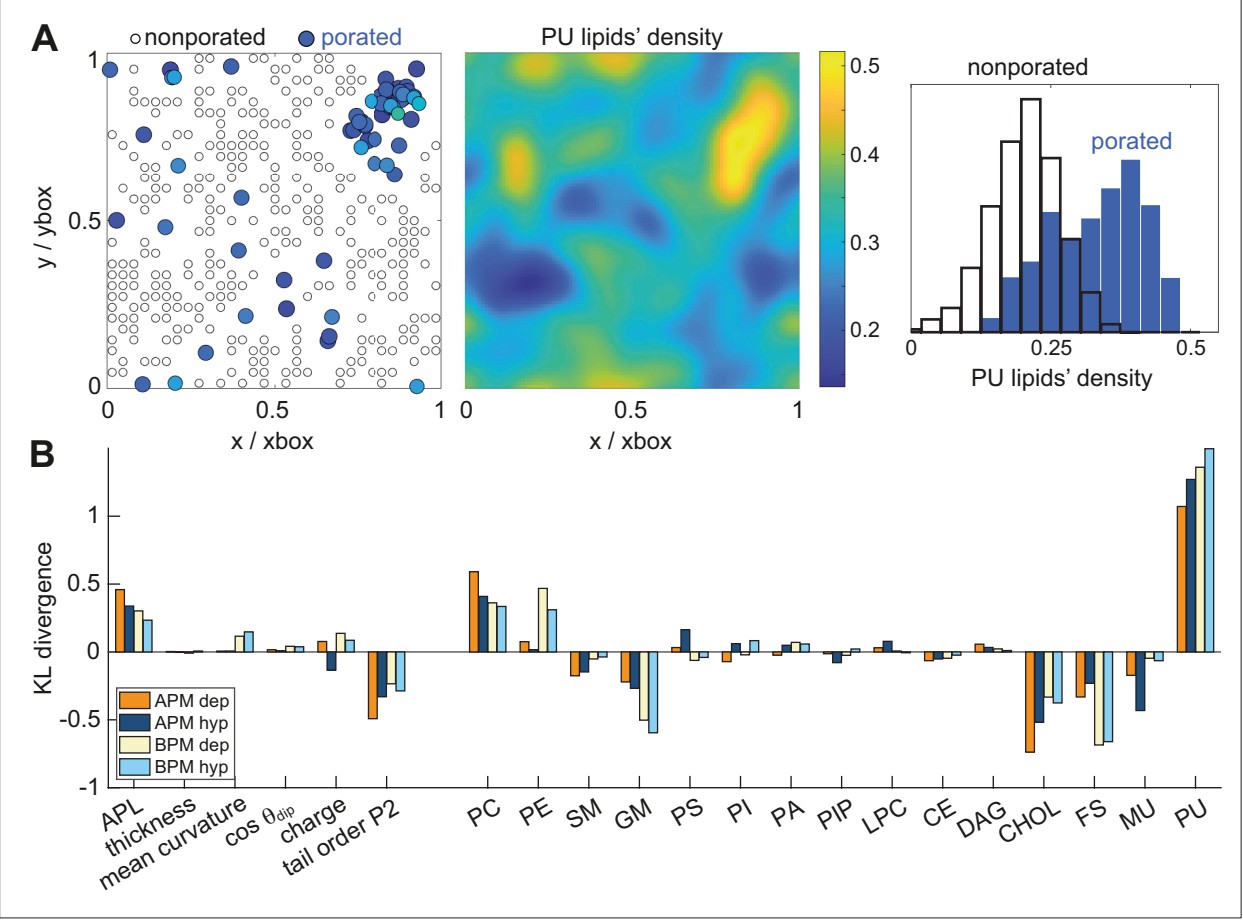

**Figure 3.** Membrane features that favour and disfavour poration. (**A**) (*Left*) Definition of porated and non-porated locations in one of the membranes; (*middle*) most pore locations colocalize with increased polyunsaturated lipids density; (*right*) the corresponding histogram of polyunsaturated lipids density values in porated and non-porated locations. (**B**) Distances (KL divergence) between probability density estimates of individual features in porated and non-porated locations. The higher the bar, the more the feature influences poration. Positive and negative bars show whether a feature favours or disfavours poration, respectively. APL, area per lipid; PC, phosphatidylcholine; PE, phosphatidylethanolamine; SM, sphingomyelin; GM, gangliosides; PS, phosphatidylserine; PI, phosphatidylinositol; PA, phosphatic acid; PIP, phosphatidylinositol phosphate; LPC, lysolipids; CE, ceramides; DAG, diglycerol; CHL, cholesterol; FS, fully saturated lipids; MU, monounsaturated lipids; PU, polyunsaturated lipids.

The online version of this article includes the following figure supplement(s) for figure 3:

**Figure supplement 1.** Membrane surface fitting with MemSurfer.

**Figure supplement 2.** Kullback–Leibler divergence.

before electric field application, from each leaflet separately and in points corresponding to locations where a pore has or has not formed in any of the electroporation simulations. The first and second group of points are labelled as 'porated' and 'non-porated' locations, respectively (*Figure 3A*, left). We estimated the distribution of the values of the various features in non-porated and porated locations by constructing histograms (*Supplementary file 1*) and quantified the difference between probability density estimates by means of the Kullback–Leibler divergence (*Figure 3B*). By far the most significant feature to distinguish the locations where pore formed from locations where they did not was the presence of PU lipids, in both the APM and BPM membranes. This finding is corroborated by visualizing the colocalization of pore clusters with nanodomains enriched with PU lipids (*Figure 3A*). However, the analysis further showed that both the lipid head and tail architecture influence poration, with pores being favoured in regions with higher density of phosphatidylcholine (PC) lipids and lower densities of cholesterol (CHOL), gangliosides (GM), and fully saturated (FS) lipids. In BPM, the pores are also favoured in regions with higher content of phosphatidylethanolamine (PE) lipids, where most PE lipids in BPM are polyunsaturated (*Figure 1—figure supplement 1*). In addition, the analysis

**Table 3.** Prediction accuracy by machine learning models, reported for the training and test datasets; dep, depolarization; hyp, hyperpolarization.

| # | Training dataset | Test dataset | Random forest all features | | SVM all features | | Neural network all features | | Random forest four features | |
|---|---|---|---|---|---|---|---|---|---|---|
| | | | Train | Test | Train | Test | Train | Test | Train | Test |
| | Same membrane composition, same *E*-field polarity; train on dataset from two out of four membranes | | | | | | | | | |
| 1 | APM-dep, mem 1 and 2 | APM-dep, mem 3 and 4 | 100% | 83% | 97% | 83% | 97% | 81% | 100% | 82% |
| 2 | APM-dep, mem 1 and 3 | APM-dep, mem 2 and 4 | 100% | 80% | 96% | 80% | 94% | 77% | 100% | 76% |
| 3 | APM-dep, mem 1 and 4 | APM-dep, mem 2 and 3 | 100% | 85% | 96% | 85% | 91% | 81% | 100% | 82% |
| 4 | BPM-dep, mem 1 and 2 | BPM-dep, mem 3 and 4 | 100% | 87% | 97% | 86% | 94% | 86% | 100% | 84% |
| 5 | BPM-dep, mem 1 and 3 | BPM-dep, mem 2 and 4 | 100% | 82% | 97% | 84% | 97% | 79% | 100% | 85% |
| 6 | BPM-dep, mem 1 and 4 | BPM-dep, mem 2 and 3 | 100% | 75% | 97% | 83% | 96% | 77% | 100% | 82% |
| | Different membrane composition, same *E*-field polarity | | | | | | | | | |
| 7 | APM-dep | BPM-dep | 100% | 83% | 94% | 82% | 93% | 83% | 100% | 83% |
| 8 | APM-hyp | BPM-hyp | 100% | 83% | 94% | 82% | 95% | 83% | 100% | 83% |
| | Same membrane composition, different *E*-field polarity | | | | | | | | | |
| 9 | APM-dep | APM-hyp | 100% | 86% | 95% | 83% | 94% | 81% | 100% | 84% |
| 10 | BPM-dep | BPM-hyp | 100% | 90% | 95% | 86% | 95% | 84% | 100% | 88% |
| 11 | APM and BPM-dep | APM and BPM-hyp | 100% | 88% | 94% | 84% | 93% | 82% | 100% | 86% |
| | Same membrane composition, same *E*-field polarity; train on 60% dataset from all four membranes | | | | | | | | | |
| 12 | APM-dep (60%) | APM-dep (40%) | 100% | 99% | 94% | 91% | 91% | 91% | 100% | 92% |
| 13 | APM-hyp (60%) | APM-hyp (40%) | 100% | 99% | 94% | 90% | 92% | 92% | 100% | 91% |
| 14 | BPM-dep (60%) | BPM-dep (40%) | 100% | 99% | 95% | 93% | 94% | 94% | 100% | 93% |
| 15 | BPM-hyp (60%) | BPM-hyp (40%) | 100% | 99% | 94% | 91% | 95% | 94% | 100% | 93% |
| 16 | APM and BPM-dep (60%) | APM and BPM-dep (40%) | 100% | 99% | 94% | 92% | 92% | 92% | 100% | 92% |
| 17 | APM and BPM-hyp (60%) | APM and BPM-hyp (40%) | 100% | 99% | 92% | 92% | 91% | 91% | 100% | 92% |

showed that pores in all membranes are favoured in regions with greater area per lipid and lower lipid order. None of the analysed features appeared markedly dependent on the electric field polarity, even when we contrasted them with 10-ns-long trajectories obtained under non-porating electric field (*Figure 3—figure supplement 2*).

### Knowing the lipid distribution is sufficient for machine learning models to reliably predict poration sites

The local poration propensity in both APM and BPM and under both electric field polarities is governed by similar features, for example, high density of PU lipids, as suggested by Kullback–Leibler divergence (*Figure 3B*). To corroborate this finding further, we trained three machine learning models, namely random forest, support vector machine (SVM), and multilayer perceptron neural network (*Fleetwood et al., 2020*), on selected subsets of data, that is, using features from APM or BPM and/or features obtained under hyperpolarization or depolarization. The accuracy of predicting poration in another/ different subset of data typically surpassed 80% in all three models tested, with random forest exhibiting slightly superior performance (*Table 3*). Similar accuracy was obtained regardless of whether we trained and tested the models on (1) datasets from different membrane pairs of the same composition and the same electric field polarity (rows 1–6 in *Table 3*); (2) membranes of the same composition but different electric field polarity (rows 7–8 in *Table 3*); or (3) membranes of different composition (rows 9–11 in *Table 3*). Visual inspection of the prediction showed that locations corresponding to pore clusters are reliably predicted, whereas those that are scattered away from the clusters tend not to be predicted as accurately (*Figure 4A*).

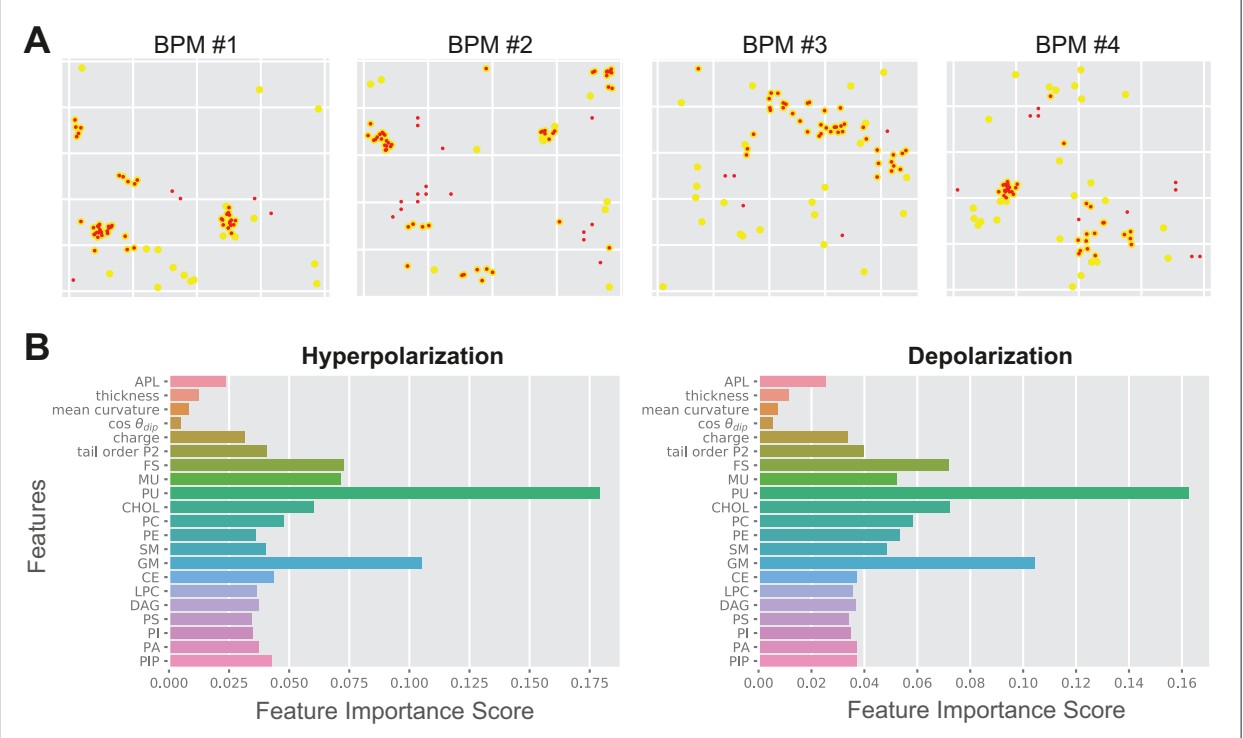

**Figure 4.** Random forest output. (**A**) Comparison between the predicted poration sites (red dots) and real poration sites (yellow dots). Only locations that were classified as porated in >90% of the trajectory frames are shown. (**B**) Feature importance score for average plasma membrane (APM) and brain plasma membrane (BPM) obtained under hyperpolarization and depolarization. The feature importance score quantifies how much each feature is used in each tree of the random forest.

The prediction accuracy became considerably improved when a single dataset was randomly split into 60% training/40% test dataset (rows 12–17 in *Table 3*). In such case, random forest reached the highest accuracy of 99%. The improvement in accuracy suggests that there is something unique about poration of each membrane, even when membranes have practically identical overall composition, which the machine learning models can capture. However, this high accuracy could be partially biased by oversampling, which was performed to balance the starting number of porated locations (~60) and the number of non-porated locations (300), see Machine learning methods. When randomly choosing 60% of a given dataset for training, the values of features from all (not just 60%) actual porated locations are effectively taken into account. Therefore, we made additional tests, where we trained random forest on the first 32 porated and 150 non-porated locations for each of the APM membrane under depolarization. The accuracy of the prediction for the rest of the porated and non-porated locations was 92%, 97%, 94%, 95% for membranes 1–4, respectively. The accuracy was lower than 99%, but still considerably higher than when training/testing on pairs of different APM membranes (rows 1–3 in *Table 3*). Consequently, this exercise suggests that prediction accuracy might be improved by finetuning machine learning models on data from more different membranes or by adding additional features to the analysis in future studies.

From the datasets for both APM and BPM, we determined the feature importance score, which confirmed that the density of PU lipids is the most important for poration, followed by density of GM, CHOL, and FS lipids (*Figure 4B*). Furthermore, the feature importance score suggested that knowing the distribution of these lipid groups is sufficient to predict poration sites. The densities of other lipids and other membrane properties including the area per lipid, thickness, …, and lipid tail order can be practically neglected. We confirmed this by training random forest using four most important features, that is the density of PU, GM, CHOL, and FS lipids. The accuracy of the prediction was indeed very similar (≥76%) as when using all features (*Table 3*, last column).

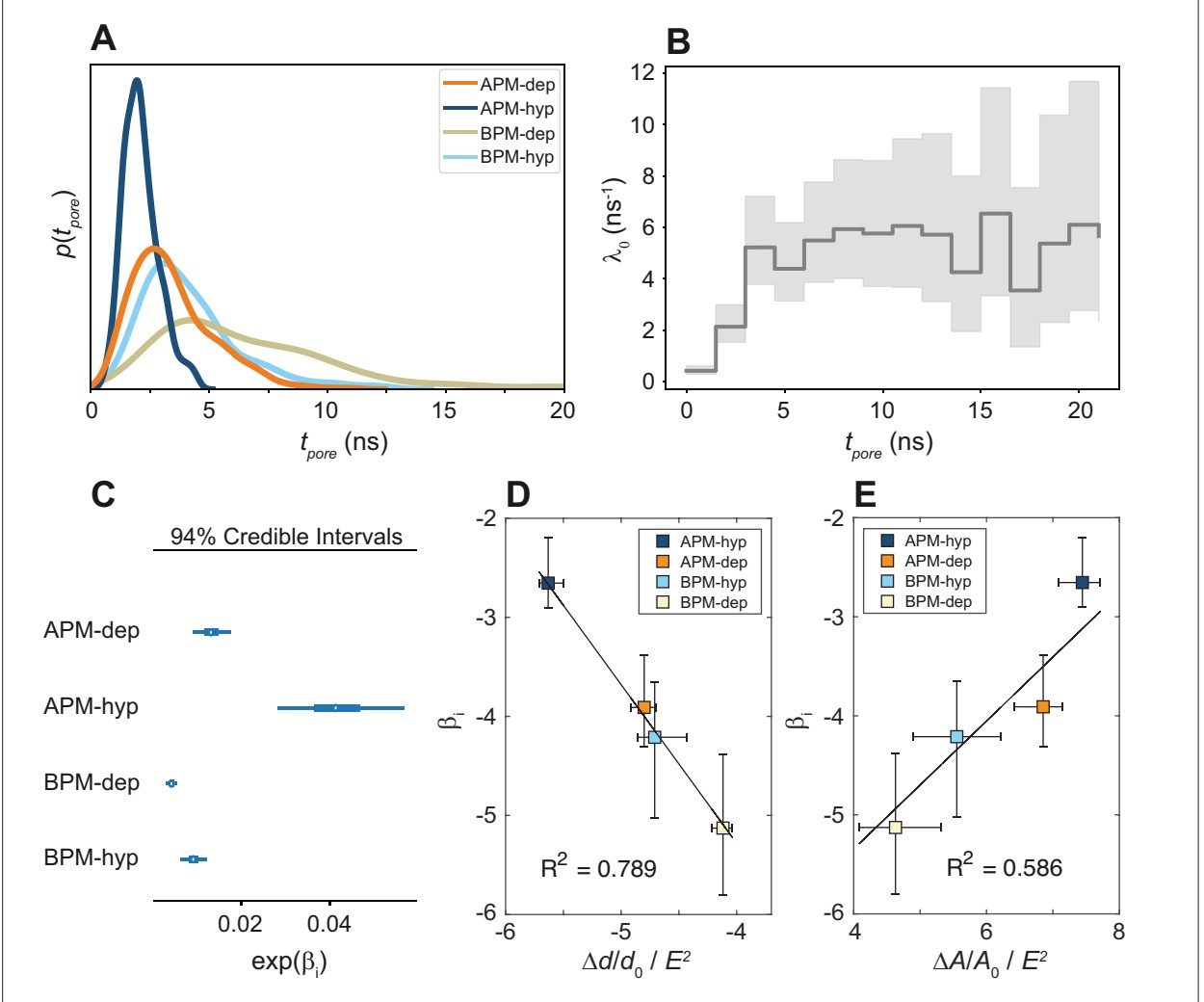

**Figure 5.** Kinetics of membrane poration. (**A**) Probability distribution of the first poration times, sampled from 60 simulations, and approximated with kernel density estimation. (**B**) The time course of $\lambda_0(t)$. (**C**) Credible intervals of $\exp(\beta_i)$ for average plasma membrane (APM) and brain plasma membrane (BPM) under depolarization and hyperpolarization. (**D, E**) Correlation between $\beta_i$ and the steady-state relative change in membrane thickness $\Delta d/d_0$ and projected area $\Delta A/A_0$ normalized by $E^2$, extracted from least-square fits presented in *Figure 5—figure supplements 5 and 6*. Square symbols show the average over all four membranes, whereas the error bars show the range of values in four membranes. The ratio $\Delta A/A_0 / E^2$ is inversely proportional to the area compressibility (or stretching) modulus (***Needham and Hochmuth, 1989***).

The online version of this article includes the following figure supplement(s) for figure 5:

**Figure supplement 1.** Posterior predictive checks of the Bayesian inference procedures.

**Figure supplement 2.** The time-dependent $\beta_i(t)$ model.

**Figure supplement 3.** Credible intervals of $\exp(\beta_i)$ separately for each of the eight membranes considered and both electric field polarities.

**Figure supplement 4.** Upon exposure to electric field, the membranes thin and increase their area.

**Figure supplement 5.** Relative change in membrane thickness under non-porating electric fields.

**Figure supplement 6.** Relative change in the projected membrane area under non-porating electric fields.

## Electroporation kinetics depends on membrane composition and electric field polarity but can be described with a universal model

The analysis presented so far demonstrates that the pore location mainly depends on the local lipid arrangement, particularly the density of PU lipids, regardless of the type of membrane and electric field polarity. However, this does not necessarily mean that poration is equally likely or equally fast in all membranes. To investigate whether the membranes differ in poration kinetics we compared the

distributions of first poration times for each membrane. These distributions were indeed found to depend both on membrane composition and electric field polarity (*Figure 5A*). APM membranes tend to porate faster than BPM, and hyperpolarizing fields tend to porate faster than depolarizing fields.

A Bayesian survival model was then trained (1) to quantify and study the statistical significance of the differences in poration rates, (2) to investigate if an underlying universal model can be used to describe the poration kinetics, and (3) to parametrize the kinetic parameters. Cox's proportional hazards survival model (*Cox, 1972*) is a suitable model for this purpose, since it allows for an arbitrary time dependence of the poration rate, and it allows to then compare different systems or situations. This model assumes that for a given system, *i*, the instantaneous probability of an event happening, $\lambda_i(t)$ (the rate pore formation), is a function of time:

$$\lambda_i(t) = \lambda_0(t) \exp(\beta_i) \tag{1}$$

where $\lambda_0(t)$ is a time-dependent and system-independent baseline rate of pore formation and $\beta_i$ are regression coefficients accounting for the difference in poration kinetics between systems (APM-depolarization, APM-hyperpolarization, BPM-depolarization, BPM-hyperpolarization). The baseline rate captures the common time dependency whereas the regression coefficient captures the system specificity. The inferred model was able to successfully reproduce the measured data (*Figure 5— figure supplement 1*) highlighting its robustness and suggesting the existence of a common kinetic model for electroporation. To further test the model validity, we trained a more complex model which allowed the $\beta_i$ coefficients to vary with time, allowing for a system-specific time variability of $\lambda_i(t)$. The $\beta_i(t)$ obtained were found to slightly drop in the interval $0 < t < 3$ ns until becoming practically time independent (*Figure 5—figure supplement 2*). Given that the $\beta_i(t)$ for all systems follow the same trend and that the time-dependent model only adds information in the initial stage of electroporation and has a high uncertainty, the time-independent model was selected for its easier interpretability and lower complexity.

The time-independent Bayesian survival model shows that the baseline poration rate $\lambda_0(t)$ is initially zero and then increases, reaching a constant steady-state value after approximately 3 ns (*Figure 5B*). To determine the mechanistic basis for the initial transient kinetic regime, we performed additional simulations at non-porating electric fields. These simulations revealed that the electric field *E* causes the membrane to thin and expand its area, whereby the relative change in membrane thickness and area are proportional to $E^2$ (*Figure 5—figure supplement 5* and *Figure 5—figure supplement 6*), as expected for Maxwell stress (*Lewis, 2003*). The membrane thickness and area reach a steady state ~3 ns after electric field onset (*Figure 5—figure supplement 4*), suggesting that the time scale of the initial transient kinetic regime is related to the progressive thinning and area expansion caused by Maxwell stress.

The calculated $\beta_i$ credible intervals (*Figure 5C*, *Figure 5—figure supplement 3*) confirm that the differences in poration kinetics between different systems are statistically significant. APM membranes porate ~5 and ~2 times faster than BPM for hyperpolarizing and depolarizing fields, respectively. Hyperpolarizing fields porate membranes ~3 and ~2 times faster than depolarizing fields for APM and BPM membranes, respectively. We found that the values of $\beta_i$ correlate with the ability of a membrane to thin and expand its area under electric field, that is, they corelate inversely with the membrane area compressibility (or stretching) modulus (*Figure 5D, E*).

## Discussion

In this study, we focused on lipid pores formed by electric field (electroporation) and we asked three main questions: Where do pores form in membranes with realistic plasma membrane lipid composition? Which membrane features/properties govern the most likely poration sites? Which membrane features/properties govern the poration kinetics?

### Features important for membrane poration

Research conducted so far has mainly focused on studying formation of pores in model membrane systems with up to three different lipid types. In membranes containing a single type of fluid-phase lipids, poration was found to depend on both the head and tail architecture. Longer tail length, larger headgroups (e.g. heads containing sugar moieties), and stronger intramolecular interactions between

both lipid tails and heads generally reduce the propensity for poration (*Ziegler and Vernier, 2008*; *Piggot et al., 2011*; *Polak et al., 2013*; *Polak et al., 2014*; *Gurtovenko and Lyulina, 2014*). Poration was further found to be more difficult in membranes with gel-phase lipids compared fluid-phase lipids (*Knorr et al., 2010*; *Majhi et al., 2015*; *Liu et al., 2014*). In binary mixtures of fluid- and gel-phase lipids, experiments suggested that pores form in the fluid domains (*Perrier et al., 2018*). Mixing a lipid with cholesterol was found to either decrease or increase the poration propensity, whereby this appears to depend on the cholesterol concentration and the architecture of the lipid (*Portet and Dimova, 2010*; *Fernández et al., 2010*; *Mauroy et al., 2015*; *Casciola et al., 2014*; *Perrier et al., 2017*). Studies were also done on ternary mixtures containing cholesterol, where the lipids organize in the liquid-ordered and -disordered domains. Both molecular dynamics simulations and experiments showed that pores preferentially form in liquid-disordered domains (*Reigada, 2014*; *Sengel and Wallace, 2016*). Overall, studies on simple model systems showed that there are many different factors influencing poration and it is quite impossible to predict which ones will be most important in a complex mixture such as the plasma membrane.

Our findings from membranes composed of >60 different lipid types mimicking the realistic composition of plasma membranes are fully in agreement with findings from simpler model systems. We observed that pores form preferentially in locations with lower lipid tail order, and that pores avoid regions enriched with gangliosides containing large sugar headgroups, regions enriched with cholesterol and fully saturated lipids. A new finding from our study is that, in plasma membranes, the most important factor governing poration propensity is the local density of polyunsaturated lipids, regardless of the membrane type (average or brain plasma membrane) and polarity of the electric field. Note that the majority of polyunsaturated lipids in both average and brain plasma membranes have either PC or PE headgroup (>75% of all polyunsaturated lipids; *Figure 1—figure supplement 1*), whereby PC and PE headgroups also contribute favourably to poration (*Figure 3B*).

## Polyunsaturated lipids and lipid oxidation

The finding that poration is strongly facilitated in the presence of polyunsaturated lipids is very important from the standpoint of lipid oxidation. Polyunsaturated lipids are highly prone for oxidation by free radicals, whereby upon oxidation a part of the lipid tail becomes hydrophilic making the membrane considerably more permeable to ions and hydrophilic molecules (*Yin et al., 2011*; *Boonnoy et al., 2015*; *Rems et al., 2019*). Several experimental studies have shown that electroporation is associated with oxidative lipid damage, and it is believed that lipid oxidation plays an important role in increased membrane permeability after electric field exposure (*Maccarrone et al., 1995*; *Breton and Mir, 2018*). Our study suggests that most pores form in domains enriched with polyunsaturated lipids. If pores somehow act as precursors for lipid oxidation, such as by improving the access for free radicals to lipid tails, this could provide a clue on how electroporation is related to lipid oxidation.

## Electroporation kinetics

Bayesian survival analysis demonstrated that all membranes are characterized by two distinct kinetic regimes. Initially, no poration occurs, but progressively as the membrane area expands due to Maxwell stress, poration becomes easier and the poration rate increases, plateauing at a maximum rate when the membrane has reached its steady-state area. The steady-state poration rate was found to be inversely correlated with membrane area compressibility modulus – a measure of how resistant the membrane is to compression or expansion (*Figure 5E*). Inverse correlation between electroporation propensity and membrane area compressibility modulus has also been found experimentally (*Needham and Hochmuth, 1989*). The increase in membrane area is directly related to a decrease in membrane thickness (*Figure 5—figure supplement 4*). Membrane thinning is expected to facilitate pore formation, as water molecules need to travel shorter distance when bridging the membrane. Note that when comparing properties in porated and non-porated locations (*Figure 3B*) the variation in local membrane thickness was not large enough to discriminate porated locations. However, pore formation was indeed found to be favoured in regions with larger area per lipid.

In the steady-state kinetic regime, the poration rate becomes constant and the time distribution of the poration events becomes exponential. A constant poration rate and an exponential distribution of first poration times are characteristic of a Poisson behaviour of the number of pores formed per unit

time. Indeed, the most accepted kinetic models of electroporation assume a constant poration rate at a fixed value of the transmembrane voltage (or the electric field within the membrane) (**Neu and Krassowska, 1999**; **Vasilkoski et al., 2006**), which is consistent with our model in the steady-state regime. Observing the non-steady-state transient kinetic regimes is beyond the time resolution of most experiments. As such, our model has the advantage of characterizing the transient initial kinetic regime, which should be important when exposing cells to increasingly used pulses with duration in the (sub)nanosecond range (**Pakhomov and Pakhomova, 2020**; **Neuber et al., 2019**).

Electroporation rate on the whole-cell level is typically modelled as (**Neu and Krassowska, 1999**; **Vasilkoski et al., 2006**; **Glaser et al., 1988**):

$$\lambda = A \exp\left(-\frac{\delta - B\Delta\Psi^2}{kT}\right) \tag{2}$$

where $\Delta\Psi$ is the transmembrane voltage, $\delta$ is the energy barrier for pore formation at $\Delta\Psi = 0$ V, $B$ is a proportionality constant, $k$ is the Boltzmann constant, and $T$ is the temperature. $A$ is a prefactor that is proportional to the number of possible pore nucleation sites and the frequency of lateral lipid fluctuations. Comparing the ratios of the pore formation rates of two systems $i$ and $j$ in **Equations (1-2)**, assuming $A_i \approx A_j$ and a steady-state kinetic regime we obtain:

$$\beta_i - \beta_j \approx -\frac{\delta_i - B_i\Delta\Psi^2}{kT} + \frac{\delta_j - B_j\Delta\Psi^2}{kT} \tag{3}$$

Therefore, the difference between $\beta_i$ of two systems is approximately equal to the negative difference of their steady-state poration-free energy barrier in $kT$ units. This relates the $\beta_i$ regression coefficients to their physical interpretation. Assuming approximately equal prefactors $A_i \approx A_j$ is reasonable for our membranes because all membranes have the same total area. In addition, the sites with the highest poration propensity are comprised of similar lipids, so we can safely assume that those regions have similar lipid fluctuation frequency. Note that the pore formation barrier $\delta$ can be independently determined by free energy methods (**Hu et al., 2015**), whereas the parameter $B$ can be inferred from simulating electroporation at different values of the transmembrane voltage; however, both these approaches are computationally more demanding. Bayesian survival analysis thus offers an alternative and simpler way to obtain parameters of **Equation 2** for different systems.

## Kinetic differences between the average and brain plasma membranes

The average plasma membranes (APMs) exhibit shorter poration times compared to the brain plasma membranes (BPMs). Bayesian survival analysis confirmed that the poration kinetics in APMs and BPMs is statistically different. Compared with BPMs, APMs contain a smaller fraction of cholesterol and fully saturated lipids and a greater fraction of polyunsaturated lipids, all favouring poration and likely increasing the poration rate. Note that the polyunsaturated lipids in BPMs have on average greater number of double bonds than in APMs (**Ingólfsson et al., 2017**) however, this is not enough to balance the poration rate with APMs. In experimental studies, cells of different types, or even cells in different phases of the cell cycle, have been found to exhibit different electroporation thresholds (i.e., different electroporation propensities) (**Cemazar et al., 2009**; **Towhidi et al., 2008**; **Golzio et al., 2002**). According to our results, this difference in thresholds is, at least in part, related to the fact that cells of different types can have considerable differences in their lipid composition, which further changes along the cell cycle (**Zhang et al., 2017**; **Atilla-Gokcumen et al., 2014**).

## Kinetic differences between hyperpolarizing and depolarizing electric field

When a cell is exposed to an electric field, its membrane becomes hyperpolarized on the side facing the positive electrode (anode) and depolarized on the side facing the negative electrode (cathode). Experiments have shown that plasma membranes of different types of cells can become more permeabilized either on the anodic or cathodic side, whereby this asymmetry in permeabilization is still not completely understood (**Mehrle et al., 1985**; **Kinosita et al., 1992**; **Tekle et al., 1994**; **Sözer et al., 2017**). Our results suggest that the asymmetric lipid composition present in all mammalian plasma membranes favours pore formation on the anodic side, which is hyperpolarized. We found that hyperpolarization induces more profound membrane thinning compared to depolarization, consequently

increasing the poration rate. Greater membrane thinning could be associated with electrophoretic drag of negatively charged lipids, which are mainly present in the inner leaflet and are pulled towards the membrane interior by hyperpolarizing electric field. Moreover, in our previous study, in which we characterized pores that formed in voltage sensors of sodium voltage gated channels (*Rems et al., 2020*), we also observed that such pores are more easily formed under hyperpolarization, albeit this was associated with asymmetric distribution of charged protein residues. Nevertheless, there are numerous types of membrane proteins in the plasma membrane, some of which might become denaturated more easily under depolarization. Whether the cell is permeabilized more on the anodic or cathodic side might therefore depend on the plasma membrane's lipid–protein content and the preferential sites of pore formation. This exemplifies the need to identify preferential poration sites in membranes with complex organization, which we discuss further in Towards building accurate cell-level models of electroporation.

## Towards building accurate cell-level models of electroporation

The cell membrane is a complex organization of lipids and proteins, whereby insights from atomistic molecular dynamics simulations suggest that the electric field can form pores both in the lipid domains and within some membrane proteins. The limitation of molecular dynamics simulations is that they are only able to model a small part of the membrane, and that they are not able to take into account the dynamic changes in the transmembrane voltage, which are present when electroporating whole cells (*Hibino et al., 1993*; *Frey et al., 2006*). Namely, on the whole-cell level, the induced transmembrane voltage varies with position on the membrane and depends on the spatiotemporal profile of the membrane conductivity. As sufficient number of pores form in the membrane, they increase the membrane conductivity to the extent that starts decreasing the transmembrane voltage and prevents formation of additional pores. In other words, in molecular dynamics simulations it is possible to observe poration of practically any membrane; however, in a real cell membrane only the sites with the highest poration propensity can be porated.

To understand electroporation of living cells, we need to be able to develop equations describing the poration rate of different types of pores (pores in the lipid domain, different membrane proteins) and embed them into a system of ordinary and partial differential equations that describe electroporation on the whole-cell level (*DeBruin and Krassowska, 1999*; *Smith et al., 2014*). Such models can then be used to study the increase in membrane conductivity, transmembrane molecular transport of different types of molecules, and changes in the membrane resting potential and/or action potentials induced by different parameters of electric pulses. Electroporation can be caried out with different pulse waveforms, where the duration of individual pulses can range from a few 100 picoseconds to tens of milliseconds. Exploring the pulse-parameter space in silico instead of through trial-and-error experimental approaches will facilitate optimization of electroporation-based applications in vitro and in vivo (*Rems and Miklavčič, 2016*).

We envision that by combining atomistic and coarse-grained molecular dynamics simulations, machine learning methods and Bayesian survival analysis we will be able to improve existing kinetic models describing electroporation on the whole-cell level. Coarse-grained simulations, together with more mesoscopic models and experiments, are in the future anticipated to enable modelling of whole plasma membranes providing their detailed molecular organization (*Pezeshkian and Marrink, 2021*). An exciting finding from our study is that knowing the lipid distribution is sufficient for identifying the most likely poration sites in lipid domains by using machine learning methods. As such, we anticipate that we will be able to use machine learning to estimate the membrane area, which is most amendable to poration, and hence estimate the prefactors in *Equation 2*. By performing electroporation simulations on selected membrane regions and applying Bayesian survival analysis, we can characterize the poration kinetics and simplify the determination of the corresponding kinetic parameters. For example, by following these approaches using coarse-grained membranes associated with actin filaments (*Schroer et al., 2020*) we can investigate and quantify how the presence of actin cytoskeleton influences poration kinetics, either by affecting the local distribution of lipids or by influencing the mechanical properties of the membrane or both. To enable finer decomposition of the relative importance of local vs. global membrane properties, future studies could combine local lipid neighborhood analysis with pore initiation rate analysis.

Coarse-grained simulations have their disadvantages. At present, coarse-grained simulations cannot be used to study poration of membrane proteins, as the protein secondary structure typically needs to be constrained for the protein to remain stable (*Periole et al., 2009*). Nevertheless, poration of membrane proteins and its corresponding kinetics can be inferred from atomistic molecular dynamics simulations (*Rems et al., 2020*). Furthermore, coarse-grained lipid bilayers are known to be more difficult to porate than corresponding atomistic bilayers, likely due to coarse-graining of multiple water molecules into a single particle (*Hu et al., 2015*). Despite being more difficult to porate, coarse-grained systems are able to represent the differences in the energy barriers for pore formation in bilayers composed of different lipid types as well as the influence of the membrane mechanical properties on poration (*Hu et al., 2015*). This confirms that we can use coarse-grained simulations to identify the lipid domains, which are the most likely to be porated. By backmapping (*Wassenaar et al., 2014*), these regions to atomistic representation, we should be able to obtain a more accurate estimation of the kinetic parameters of *Equation 2*. Nevertheless, we hope our study will motivate further exploration on the validity of coarse-grained membrane models for studying membrane electroporation, both in comparison with corresponding atomistic computational membrane models and experimental model membrane systems.

## Conclusions

Pores in the plasma membrane can be formed under diverse physicochemical conditions. They can be formed in various physiological processes by pore-forming proteins, and when the membrane is subject to external mechanical or electromagnetic forces (*Gilbert et al., 2014*; *Sun et al., 2020*). In this study, we investigated pores formed by electric field. Electroporation simulations of coarse-grained membranes mimicking realistic lipid composition of plasma membranes showed that pores preferentially form in domains enriched with polyunsaturated lipids and that pores avoid domains enriched with gangliosides, cholesterol, and fully saturated lipids. The density of polyunsaturated lipids is the most important feature governing the preferential pore location, regardless of the overall membrane composition and electric field polarity, as corroborated by machine learning methods. However, the poration kinetics does depend significantly on membrane composition and electric field polarity, as demonstrated by Bayesian survival analysis. The poration rate is higher under hyperpolarizing compared to depolarizing electric field and correlates with the ability of a membrane to expand its area under electric field. We envision that by combining atomistic and coarse-grained molecular dynamics simulations, Bayesian survival analysis and machine learning models, we will be able to improve existing kinetic models describing electroporation on the whole-cell level. Although we have focused on pores induced by electric fields, the findings are likely to be applicable to other ways of membrane poration, as we found that the lipid organization is much more important for poration than the electrical features of the membrane (charge, dipole angle).

## Materials and methods
### Molecular dynamics simulations
### System preparation

The starting point for our systems was the topology files for the coarse-grained membranes used in the study of *Ingólfsson et al., 2017*, available at https://bbs.llnl.gov/neuronal-membrane-data.html. The membranes are parametrized with the Martini 2.2 force field (*Marrink et al., 2007*; *de Jong et al., 2013*). We took the frames extracted after 80 µs of the simulation (the files confout-80us.gro). The original membranes were about 70 nm × 70 nm large. We cut four 30 nm × 30 nm pieces from the original membranes. After cutting, we removed an appropriate number of Na or Cl ions, such that the final system had zero net charge. The NaCl concentration was ~150 mM. We replaced the non-polarizable water model with its polarizable version to have a more accurate system representation for electroporation studies (*Yesylevskyy et al., 2010*). We also added more water to each system, such that the simulation box in *z* direction was ~19 nm after equilibration. This procedure was done using functions from VMD (Visual Molecular Dynamics) (*Humphrey et al., 1996*), Gromacs (*Abraham et al., 2015*), and custom scripts. In the end we had eight membrane systems, four with lipid composition corresponding to the APM and four corresponding to the BPM composition.

## Equilibration and production runs

After preparing the new systems and topology files, we made two steps of system minimization and four steps of system equilibration following the equilibration protocol for Martini membranes from charmm-gui.org (*Qi et al., 2015*). We then ran a short 500 ns simulation using the reaction-field electrostatics followed by 50 ns equilibration using Particle Mesh Ewald (PME) electrostatics (*Darden et al., 1993*). Some of the PC lipid's heads were constrained in their $z$ position to reduce bilayer undulations, as done in the original publication (*Ingólfsson et al., 2017*). Other MD parameters were equal to the default parameters for simulations with Martini membranes from charmm-gui.org in April 2020: leap-frog integration of the equations of motion using 20 fs time step; plain cutoff of van der Waals and Coulomb interactions at distance 1.1 nm; relative dielectric constant $\varepsilon_r$ = 2.5; temperature coupling using velocity rescaling with a stochastic term (*Bussi et al., 2007*) with time constant 1 ps and temperature of 310 K; semi-isotropic pressure coupling using Parrinello–Rahman barostat (*Parrinello and Rahman, 1981*) with time constant 12 ps and reference pressure of 1 bar, with the compressibility of the system set to 3e−4 bar$^{-1}$. The pressure coupling allows the size of the simulation box to adjusts to the changes in the membrane area. The trajectory was saved every 0.1 ns. All simulations were carried out with Gromacs 2019.4 (*Abraham et al., 2015*).

## Electroporation simulations

After equilibration, and for each of the eight membranes, we ran multiple replicas of electroporation simulations where the membranes were exposed to an electric field of +127.7 mV/nm (60 replicas) or −127.7 mV/nm (60 replicas). No positional restraints were imposed in these simulations. The electric field was chosen such that we observed the formation of at least one pore within ~15 ns. Having a short poration time was important, because we aimed to map the local membrane features before poration to the likelihood of a poration event, which means that we needed to avoid considerable lateral lipid diffusion. The value of the electric field $E$ imposed in simulations, and reported above, cannot be directly compared to that reported in experiments. The reason for this is the manner in which $E$ is implemented in molecular dynamics. Namely, each particle carrying a charge $q_i$ is assigned an additional force $F = q_i E$. As such, $E$ corresponds to the electric field strength that would exist in vacuum. However, in our molecular dynamics systems there are many electric dipoles, including water molecules and headgroups of zwitterionic lipids, which respond to $E$ by changing their average orientation, that is, they polarize. The ability of a material to polarize under electric field is characterized by relative dielectric permittivity, which corresponds to the factor by which the electric dipoles within the material reduce the electric field that would exist in vacuum. Since the relative dielectric permittivity of water is around 80 (both experimentally and for the polarizable MARTINI water model *Yesylevskyy et al., 2010*), the macroscopic electric field that establishes in the aqueous compartment of our molecular dynamics systems is about two orders of magnitude lower compared to the imposed electric field $E$ (*Vernier et al., 2013*), that is, on the order of 1 mV/nm = 10 kV/cm. This macroscopic electric field strength is typically reported in experiments after being determined, for example, as the voltage-to-distance ratio, if placing the sample between a pair of parallel-plate electrodes. The macroscopic electric field strengths in our simulations are well within the range of those used in experiments, when exposing cells to submicrosecond pulses (order of 1–100 kV/cm). Furthermore, the probability of pore formation depends exponentially on the magnitude of the applied electric field (*Equation 2*; *Böckmann et al., 2008*). The electric field strength required for detectable electroporation thus reduces with increasing the duration of the applied electric pulse, and consequently macroscopic electric field strengths of the order of 0.1–1 kV/cm are typically sufficient for electroporation when exposing cells to conventional microsecond and millisecond electric pulses. While the conditions in our study strictly correspond to nanoseconds-long exposures to electric field, we expect similar behaviour also for longer electric pulses. Nevertheless, on a longer time scale, electrodeformation of the cell membrane and/or increases in local membrane curvature caused by electric field might play additional roles in the pore formation process (*Perrier et al., 2017*; *Riske and Dimova, 2005*).

## Additional simulations under non-porating electric field

For each membrane, we also ran five 10-ns-long simulations under lower electric field strengths of 0, ±42.5, ±63.8, ±85.1, and ±106.3 mV/nm. No positional restraints were imposed in these simulations. The initial coordinates (initial gro file) were the same as for electroporation simulations. With the

exception of a few simulations at ±106.3 mV/nm, these electric field strengths were too low to induce poration within 10 ns.

## Pore localization

To determine the poration time and pore location in the electroporation trajectories we used a custom semi-automatic procedure with Python and MDAnalysis (https://www.mdanalysis.org/; *Michaud-Agrawal et al., 2011*). The procedure consisted of three main steps, as described below. For analysis, we generally considered only the first poration event. Additional pores could form after the first one and all pores eventually expanded until destroying the membrane, as is usual for simulations under constant electric field (*Fernández et al., 2012*; *Delemotte and Tarek, 2012*). We focused on the first poration event, as the local electric field changes after poration, changing the transmembrane voltage. In up to ~20% simulations, two or more pores formed practically simultaneously. In such a case, we considered all of these pores for analysis.

### Step 1 – coarse search

Start from frame 12 (since no pore formed before time 1.2 ns), search every six frames (i.e. 12, 18, 24, …). In each frame, divide the membrane into 100 small pieces. In each small piece, look for water molecules around the centre of the lipid bilayer. If there are more than 12 water molecules, stop and write down the frame ($t$) and the region ($i,j$), otherwise, continue the loop. This step is used to speed up and improve the accuracy of the algorithm. The condition of 12 water molecules ensured that only fully formed pores were considered for the second step.

### Step 2 – precise Search

Start from frame $t - 5$, search every frame until $t$ (i.e. $t - 5$, $t - 4$, $t - 3$, …). In each frame look for water molecules within the centre of the lipid bilayer in the regions ($i - 2, j - 2$), ($i - 2, j - 1$), …, ($i + 2, j + 1$), ($i + 2, j + 2$). If there are more than four (APM) or six (BPM) water molecules, check the following two frames. If there are water molecules also in the following two frames, return the poration time and the centre of mass of these water molecules as the pore location. Otherwise, continue the loop.

### Step 3 – manual check and correction

The procedure failed for BPM, when the pore formed later than ~7 ns after the onset of the electric field, mainly because of the increase in BPM curvature with time. In this case, the poration time and location were determined by visualization of the trajectory in VMD.

All poration times and pore locations were manually verified by extracting the porated frames from all trajectories and translating the systems in ($x,y$) such that the pore location moved to a predefined position. These extracted frames were then visualized in VMD.

## Analysis of membrane properties

To determine local membrane properties we used the MemSurfer tool (*Bhatia et al., 2019*), available at https://github.com/LLNL/MemSurfer, (*Lawrence Livermore National Laboratory, 2022*; copy archived at swh:1:rev:51c7c0534f7b73e74e7233e44c08a7fb269a0188) MemSurfer is a 3D membrane surface fitting tool, which fits a surface to the inner and outer membrane leaflet and uses Delaunay triangulations and surface parameterizations to compute membrane properties of interest (*Figure 3— figure supplement 1*). Each vertex is mapped to the position of a lipid headgroup. The triangulated surface is then further smoothed (*Figure 3—figure supplement 1D*).

MemSurfer can be used to measure local membrane thickness, area per lipid, and mean curvature, among others. We added functions that analyse the dipole angle of zwitterionic lipids, the lipid tail order, and the headgroup charge. The dipole angle for each zwitterionic lipid was computed as the angle between the local membrane normal (determined by MemSurfer) and the vector connecting two beads corresponding to the phosphate group and the choline or amine group ($PO_4$ and $NC_3$ or $NH_3$). To determine lipid tail order, we first computed the angles between the local membrane normal and the vectors connecting all adjacent beads in the lipid tails. Then we computed the average of the cosine of all angles and determined the average order parameter as

$$P_2 = \tfrac{1}{2}\left(3\langle\cos\theta_i\rangle^2 - 1\right) \tag{4}$$

where the average goes over all bond angles in both lipid tails. This definition of the lipid order turned out to be more sensitive for separating values in porated and non-porated regions, compared to the more common definition $P_2 = \tfrac{1}{2}\left(3\langle\cos^2\theta_i\rangle - 1\right)$ (**Ingólfsson et al., 2017**). We also extracted the type of lipid at a given vertex and the charge of this lipid. The scripts for running MemSurfer in this study, as well as the following analysis carried out in Matlab R2021a and described below, are available at https://github.com/learems/Electroporation-CGmem-MemSurfer, (copy archived at swh:1:rev:3e7d2d393b8ec98e04aa1e8915b55ee024840b97; **Rems, 2021a**).

MemSurfer returns the values of the above-listed properties/features at vertices, which correspond to individual lipids' headgroups. For a given trajectory frame, the values of most of the properties of interest like area per lipid, lipid order parameter, charge, … can exhibit large variations among the adjacent vertices. Therefore, to extract the values at a selected $(x_p,y_p)$ position, we used a Gaussian smoothing kernel, which determined a weighted average of the values $V_i$ at all vertices within a radius $3r_{smooth}$ from the $(x_p,y_p)$ position

$$V_{smooth}\left(x_p,y_p\right) = \frac{\sum_i V_i w_i}{\sum_i w_i} \tag{5}$$

where the weights for a given adjacent vertex $i$ are

$$w_i = \exp\left(-\frac{1}{2}\frac{(x_i-x_p)^2}{r_{smooth}^2} - \frac{1}{2}\frac{(y_i-y_p)^2}{r_{smooth}^2}\right) \tag{6}$$

When determining the presence/density of lipid groups (PC, PE, SM, etc.), we assigned a value of 0 or 1 if a given lipid group was located at a given vertex. Gaussian smoothing was carried out in the same way as for other features. The same smoothing procedure was also done for plotting the surface plots in **Figure 5**. We chose $r_{smooth}$ = 1.5 nm, because this resulted in the best separation of the feature values in non-porated and porated regions.

The porated locations were defined as the locations at which a pore formed in any of the 60 electroporation simulations. For non-porated locations, we first divided the membrane into a grid with 31 by 31 points. We excluded all points which were within ~2 nm (6.7% of the box size) of any porated location. We also randomly excluded excess points such that the final number of non-porated locations was 300 (**Figure 3A**).

The distances between histograms of feature values in porated and non-porated locations (**Figure 3B**) were quantified by symmetrized version of the Kullback–Leibler divergence (**Fleetwood et al., 2020**) after performing a kernel smoothing probability density estimate:

$$distance = \pm\frac{1}{2}\int_{x_1}^{x_2}\left(p\left(x\right)\log\left(\frac{p\left(x\right)}{q\left(x\right)}\right) + q\left(x\right)\log\left(\frac{q\left(x\right)}{p\left(x\right)}\right)\right)dx \tag{7}$$

where $p(x)$ and $q(x)$ are the distributions of the data in porated and non-porated locations, respectively. The limits $x_1$ and $x_2$ were defined as the lowest and highest value of $x$, where either of the distributions fell under 1% of their peak value. In **Figure 3B**, the distances correspond to probability density estimates, which were obtained after averaging the value of a given feature and at a given point over both membrane leaflets. Distances computed for each leaflet separately are shown in **Figure 3—figure supplement 2**.

## Machine learning methods

The values of all the features extracted in non-porated and porated locations from 101 frames of a 10-ns-long trajectory before electroporation were used to train three machine learning models with Python and scikit-learn library (**Pedregosa et al., 2011**): random forest, support vector machine and multilayer perceptron neural network. The input data contained the $(x,y)$ locations on the membrane and the extracted membrane properties at those locations (i.e., features, denoted as $X$). The input data were separated into two classes with label $Y = 1$ for porated locations and $Y = 0$ for non-porated locations. As the number of non-porated locations (300) and porated locations (~60) was imbalanced, we used SMOTE in Python to randomly oversample the data in porated locations and balance them

with the data in non-porated locations. After oversampling the data, we trained the above-listed machine learning models and used them to predict the pore locations. The accuracy of the prediction was evaluated as:

$$accuracy = \frac{no.\ correctly\ predicted\ points\ (porated\ and\ nonporated)\ in\ all\ frames}{no.\ all\ points\ \times\ no.\ all\ frames} \tag{8}$$

The feature importance score was determined using the attribute feature_importances_ of the scikit-learn random forest model. The codes used for this analysis are available at https://github.com/learems/Electroporation-CGmem-MachineLearning, (copy archived at swh:1:rev:1e32cf42c04af4543f-b101ece2c92f35d793d62e; **Rems, 2021b**).

### Bayesian survival analysis

To model the poration rates, we used Bayesian survival analysis. Given the non-exponential shape of the first poration times and the heterogeneity of poration rates between systems, it was crucial to use a model that allowed for an arbitrary time dependence of the rate and that was able to account for the system dependence. Cox's proportional hazards model has these features and defines the event rate as

$$\lambda\left(t\right) = \lambda_0\left(t\right)\exp\left(\sum_i \beta_i x_i\right) \tag{9}$$

$\lambda_0(t)$ is constructed as a piecewise constant function defined in time intervals with endpoints: $0 \leq s_0 \leq \ldots \leq s_N$ such that $\lambda_0(t) = \lambda_j$ for $s_j \leq t < s_{j+1}$. $x_i$ are binary categorical variables such that if the poration time describes a system $k$, then $xi_{i=k} = 1$ and $x_{i\neq k}k_0$, that is, one-hot-encoding representations of the system category. A good signal-to-noise ratio was found with 1.5 ns time intervals. $\lambda_j$ were given independent priors in the form of gamma distributions with a shape parameter of 50 and a scale parameter of 10 ns$^{-1}$. $\beta_i$ were given normally distributed priors centred at the origin and with a standard deviation of 100. Both priors are fairly uninformative.

The Bayesian inference of the model parameters $\{\lambda_i, \beta_i\}$ was done using as input features the first poration times and the system category (APM-dep, APM-hyp, BPM-dep, BPM-hyp) encoded in $x_i$. In this way, the model was inferred on the electroporation events of all systems and a posteriori the model for a particular system $i$, $\lambda_0(t)\exp(\beta_i)$ was computed. Posterior predictive checks validated the quality of our models: the model can generate data that reproduce accurately the observed data (**Figure 5—figure supplement 1**).

To test the universality of $\lambda_0(t)$ and check the time independence of $\beta_i$, we inferred another model allowing $\beta_i$ to be time dependent:

$$\lambda\left(t\right) = \lambda_0\left(t\right)\exp\left(\sum_i \beta_i\left(t\right)x_i\right) \tag{10}$$

In this alternative model, $\beta_i(t)$ is also a piecewise constant function analogously to $\lambda_0(t)$. The heights of the steps of $\beta_{i,j}$ for $s_j \leq t < s_{j+1}$ were given gaussian random walk priors: $\beta_{i,j}$ is given a gaussian prior with standard deviation of 1 but centred on their previous $\beta_{i,j-1}$.

Since we are using Bayesian statistics, the parameters of the model $\{\lambda_j, \beta_i\}$ are treated as random variables whose distributions are inferred as a result. We can calculate their 94% credible intervals, that is, the interval containing 94% of the probability density of the variable. Therefore, uncertainties are presented as the median of the variable ± the distance to the credible interval limits.

The models were inferred using PYMC3 (**Salvatier et al., 2016**) and follow a similar methodology to one of the library's case studies (available here). The full details of the implementation are available at https://github.com/sperezconesa/electroporation_modeling, (copy archived at swh:1:rev:1c-c26d28bd76b5c254b4cbbf85703b2394aff775; **Pérez-Conesa, 2022**).

## Acknowledgements

This work was supported by grants from the Science for Life Laboratory, from the Gustaffsson foundation from Swedish Research Council (VR 2018-04905) to L.D., and by a synergy postdoc grant to L.D. and I.T. from KTH Royal Institute of Technology. The work was also supported by funding from the European Union's Horizon 2020 research and innovation programme under the Marie Skłodowska-Curie grant agreement no. 893077 to L.R., and by funding from Slovenian Research Agency (ARRS)

under project no. J2-2503. The simulations were performed on resources provided by the Swedish National Infrastructure for Computing at parallelldatorcentrum (PDC) Centre for High Performance Computing and at High Performance Computing Center North (HPC2N).

## Additional information

### Competing interests
Ilaria Testa, Lucie Delemotte: Reviewing editor, *eLife*. The other authors declare that no competing interests exist.

### Funding

| Funder | Grant reference number | Author |
|---|---|---|
| Science for Life Laboratory | | Ilaria Testa<br>Lucie Delemotte |
| Vetenskapsrådet | 2018-04905 | Lucie Delemotte |
| Gustafssons Stiftelse | | Lucie Delemotte |
| European Commission | H2020-MSCA-IF 893077 | Lea Rems |
| Slovenian Research Agency | J2-2503 | Lea Rems |

The funders had no role in study design, data collection, and interpretation, or the decision to submit the work for publication.

### Author contributions
Lea Rems, Conceptualization, Formal analysis, Funding acquisition, Investigation, Methodology, Project administration, Software, Supervision, Validation, Visualization, Writing – original draft, Writing - review and editing; Xinru Tang, Fangwei Zhao, Data curation, Formal analysis, Investigation, Visualization, Writing – original draft; Sergio Pérez-Conesa, Conceptualization, Formal analysis, Investigation, Methodology, Software, Validation, Visualization, Writing – original draft, Writing - review and editing; Ilaria Testa, Conceptualization, Funding acquisition, Investigation, Supervision, Validation, Writing - review and editing; Lucie Delemotte, Conceptualization, Funding acquisition, Investigation, Project administration, Supervision, Validation, Writing - review and editing

### Author ORCIDs
Lea Rems  http://orcid.org/0000-0001-7470-4367
Ilaria Testa  http://orcid.org/0000-0003-4005-4997
Lucie Delemotte  http://orcid.org/0000-0002-0828-3899

### Decision letter and Author response
Decision letter https://doi.org/10.7554/eLife.74773.sa1
Author response https://doi.org/10.7554/eLife.74773.sa2

## Additional files

### Supplementary files
• Transparent reporting form

• Supplementary file 1. Histograms of tested features. Histograms showing the values of tested features in non-porated (empty bars) and porated (blue bars) locations, for average plasma membrane (APM) and for brain plasma membrane (BPM). The purple and yellow solid lines show kernel smoothing density estimates. The title above each group of graphs indicates the dataset for which the histograms are shown.

### Data availability
The current manuscript is a computational study. Data required to reproduce the results is available at https://osf.io/fv98a/. Analysis codes are available at https://github.com/learems/

The following dataset was generated:

| Author(s) | Year | Dataset title | Dataset URL | Database and Identifier |
|---|---|---|---|---|
| Rems L, Perez-Conesa S, Delemotte L | 2021 | Electroporation through CG-MD, machine learning and bayesian survival analysis | https://osf.io/fv98a/ | Open Science Framework, fv98a |

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
