## [Decision Letter]

**Decision letter after peer review:**

Thank you for submitting your article "Identification of electroporation sites in the complex lipid organization of the plasma membrane" for consideration by *eLife*. Your article has been reviewed by 3 peer reviewers, and the evaluation has been overseen by Qiang Cui as Reviewing Editor and José Faraldo-Gómez as Senior Editor. The following individuals involved in review of your submission have agreed to reveal their identity: Helgi I. Ingolfsson (Reviewer #2); Rumiana Dimova (Reviewer #3).

Essential revisions:

1) Additional discussion on the magnitude of voltage/electric field relative to typical experimental values and implications on the simulation results.

2) Analysis of simpler membranes to explicitly establish key factors that dictate poration kinetics and mechanism.

The reviewers also raise a number of issues related to the validation of the coarse-grained model, presentation of technical details and wording in certain discussions.

*Reviewer #2 (Recommendations for the authors):*

I realize validation of the coarse-grained results are likely out of scope for this already quite extensive manuscript (i.e. currently a number of the lipids in the complex membranes used don't have AA force field parameters and a in-between complex mixture would have to be used, etc, etc), therefore, I tried to suggest the validation as future directions. I do have a number of more specific questions and comments listed below:

One perception and/or wording comment. The brain is commonly thought to be enriched in polyunsaturated lipids. But you write that APM has more PU than the BPM:

L100 "Compared with BPM, APM has lower fraction of cholesterol, higher fraction of phosphatidylcholine (PC) lipids and higher fraction of monounsaturated (MU) and polyunsaturated (PU) lipids in the top leaflet." and

L391 "Compared with BPMs, APMs contain a smaller fraction of cholesterol and fully saturated lipids and a greater fraction of polyunsaturated lipids,"

This looks to be correct from how you defined the PU group (including also double unsaturated tails), compared to the 0, 1, 2, 3+ grouping in Figure 1A in Ingolfsson et al., 2017 (ref 41). Also, same paper Table 1, shows that on average (for the lipid fraction) there are more unsaturations per tail in the BPM than in the APM. Maybe some explanation or redefinition could prevent confusion for future readers.

Somewhat ill-defined wording comment. When first reading the manuscript, I got the sense that the Bayesina analysis was giving different result than the lipid feature/ML analysis. After reading it more in-depth I understand that they show different results on different "things" propensity/time vs spatial location and therefore are complementary. Maybe some rewording in the manuscript can make this clearer. I think my biggest stumbling block was "however" instead of something like additionally.

P1,L22, "learning. However, by" However ….

P2 "However, by analysing poration kinetics with Bayesian survival analysis we then show that"

I really like the analysis in section 2.4, but confused by one thing. It looks like form Figure 5 – sup. 1 and Figure 5B that the model is unable to capture the initial delay in pore formation time, the initial ~2.5 ns discussed later due to the bilayer expansion. Is this my misunderstanding or due to granularity of the binned data and/or lack of x-d offset term in β?

P16, L354 "Since membranes are practically volume-incompressible, the increase in area is directly related to a decrease in thickness.". Did you check this in your case? i.e. this is true for actual bilaye area, and projected area for very small flat bilayers but not larger undulating bilayers. I know both the APM and BPM were equilibrated with Z restrains to make them mostly flat but Ingolfsson et al., 2017 Figure 4A shows they are not totally flat and the BPM has more local undulations.

P16, L355 "Membrane thinning is expected to facilitate pore formation, as water molecules need to travel shorter distance when bridging the membrane.". Yes, but is it possible to utilize the power of the simulations and check for water bridging more directly? Here I realize this might be outside the scope of this manuscript, but have you considered looking at local bilayer properties such as "defects" (water penetration to the tails) or local min water distance across the bilayer (water dependent and not lipids dependent like the very related bilayer thickness)?

P9, L215+. I was quite confused reading that paragraph (L215 to L226), granted I am not an ML expert, but maybe some further explanation is in order. Also, could the questions of overfitting vs need for more data for higher accuracy be solved by increasing the number of depolarizable/hyperpolarizable simulation repeats? and/or leaving out some data for validation checks?

*Reviewer #3 (Recommendations for the authors):*

1. One aspect that remains unclear in this work relates to how universal the reported findings are and whether the large fraction of species present in the simulations is essential to the reported outcome, especially regarding the dependence on membrane elasticity. Indeed, to establish this correlation, it would have been interesting, using the selected approaches to first explore simpler membranes, and in particular, directly compare the poration probability and kinetics as a function of composition and membrane leaflet asymmetry. Membrane compositions with only a few representatives of the major classes of components would have been a good start as this would allow direct comparison to experimental results and would be helpful to resolve the role of the main actors involved.

2. The introduction (paragraph starting at line 70) explains that experimentally, pores in lipid membranes are difficult to assess. Work on giant unilamellar vesicles showing direct imaging of pores in lipid membranes suggests the opposite and should be referenced here (see e.g. DOI: 10.1529/biophysj.104.050310 and DOI: 10.1002/advs.202004068).

3. It will be good to explicitly discuss the following: The poration of the membranes in the simulations occurs within the first 15 nanoseconds, implying that the findings apply to pulses of high amplitude and nanosecond duration, but not necessarily to pulses of lower field strength and in the micro- or millisecond range duration (i.e. including conditions used in medical applications). A comment about this is due.

4. The authors indicate that such short poration times are needed to minimize lateral diffusion and allow mapping of local membrane features. Presumably, lipid diffusion and mixing which is substantial on longer times scales, might jeopardize the validity of the findings for longer pulse duration. Could the authors introduce a discussion also about this aspect?

5. The authors should explicitly specify the solution in which the membranes are simulated and the ionic strength.

6. The finding that gangliosides are quite important for poration (Figure 4B) is very interesting. The authors should discuss possible reasons and potential implications. Along these lines (even though not directly comparable), there has been a recent report on the poration of GM1-doped vesicles which exhibit much longer pore lifetimes compared to PC membranes (DOI: 10.1073/pnas.1722320115).

7. The authors should explicitly specify whether the size of the simulation box adjusts to accommodate the area of the pores in the membrane and whether more than one pore are simultaneously detected in one membrane patch. In line 379, the authors state that one can assume that Ai ≈Aj because all membranes have the same total area. Do they refer to the area before or after the application of the field? Could the authors further clarify the connection between the number of possible pore nucleation sites (as suggested on line 373) and the total area of the membranes?

8. Similarly, how does the pore size compare to the size of the data points displayed in Figure 2. In the caption of Figure 2, to avoid confusion, the authors should specify that all points correspond to the first poration event.

9. Figure 3 —figure supplement 3: The authors state that these quantities are computed from equation (1), however the caption above the graphs indicates that these two plots correspond to data where either no field was applied or a non-porating field was applied (i.e., situations where no pore is supposed to form in the membrane). How were these two diagrams plotted as they compare non porated with porated locations? Please clarify this in the caption.

10. The data for change in the membrane area as a function of the square of the nonporating field strength (Figure 5D and supplement 4) should be discussed in terms of the stretching elasticity modulus of the membranes. Typical stretching elasticity moduli of single-lipid membranes lie around 250 mN/m (see e.g. DOI: 10.1016/S0006-3495(00)76295-3). To claim the validity of the reported correlation between poration and mechanics, it has to be clarified, whether the applied simulation force fields can be used to correctly reflect the membrane elasticity. To address this (if not already reported for the selected force fields), the authors should measure the stretching elasticity for single-lipid membranes and compare with experimental values.

11. Line 270: "is fairly time independent" should be reformulated and justified by further discussion. As the authors refer to the values of β_i of Figure 5 – Supplement figure 2, the trend in this figure shows two main problems: (i) the values are positive for the whole time interval which is inconsistent with Figure 5 showing only negative values, and (ii) the values look actually time dependent as for instance APM-hyp shows a drop of 25% of its initial value on the time interval where its related probability density displayed Figure 5 A is non zero. Please discuss this issue.

---

## [Author Response]

Essential revisions:1) Additional discussion on the magnitude of voltage/electric field relative to typical experimental values and implications on the simulation results.

This has been addressed, as described in response to comment (R1.1).

2) Analysis of simpler membranes to explicitly establish key factors that dictate poration kinetics and mechanism.

We would like to argue that key factors that dictate poration kinetics and mechanism have already been established on simpler membranes. However, from studies on simple systems it is difficult to predict what will be the key factors dictating poration in a complex lipid mixture such as the plasma membrane. Our study is specifically designed with the aim to explore poration in a complex plasma membrane mimic. In addition, we develop a methodology that will allow us to establish quantitative models for the poration kinetics in plasma membranes at the whole-cell level.

The results presented in our study (especially Figure 3B, Figure 4, and Figure 5D,E) show that the following factors influence poration:

1. Lipid headgroup architecture (PC lipids are most prone for poration and gangliosides least prone for poration)

2. Lipid tail architecture (polyunsaturated lipids are more prone for poration than mono- and fully saturated lipids)

3. Cholesterol content and lipid tail order (less cholesterol and less order more likely poration)

4. Membrane thickness (thinner membrane more likely poration)

5. Membrane area compressibility or stretching modulus *K_A_* (lower *K_A_* more likely poration)

Below we provide a brief literature review summarizing previous findings on simpler lipid systems, using molecular dynamics (MD) simulations, experiments on giant unilamellar vesicles (GUVs), or experiments on planar lipid bilayers. This literature review shows that our findings are fully in agreement with previous findings on simpler systems. As Reviewer 1 correctly pointed out, our findings of what influences poration are all somewhat expected qualitatively. However, the literature review shows that there are many different factors influencing poration and it is quite impossible to predict which ones will be most important in a complex mixture. Consequently, our study fills this knowledge gap.

Influence of lipid headgroup and tail architecture

Previous studies showed that poration propensity *in membranes containing* fluid phase lipids depends on the architecture of both the lipid head groups and tails. Polak et al., studied electroporation of monocomponent bilayers consisting of lipids with different tails and headgroups using atomistic MD simulations (Polak et al., 2014, 2013). They demonstrated a considerate effect of methyl branches in the lipid tails, as well as the type of linkage between the head group and the carbonyl region. They observed that *poration propensity decreases*, respectively, in linear-chained DPPC lipids, methyl-branched DPhPC with ester linkages, and DPhPC with ether linkages, all in the fluid phase. Based on their analysis, they proposed that the presence of methyl branches could reduce the mobility of water molecules in the hydrophobic core and hence reduce the poration propensity. Note that DPhPC lipids are not present in our average and brain plasma membranes; however, knowing the characteristics of this lipid is important for understanding certain results referenced later. Polak et al., further studied *electroporation* of archaeal lipids, which have the same tail structure as DPhPC-ether lipids, whereas the archaeal head groups are formed by large sugar moieties (Polak et al., 2014). Compared with DPhPC-ether, archaeal lipids are more difficult to porate, associated with stronger interactions between the archaeal head groups. Similarly, Piggot et al., found that outer membranes of Gram-negative bacteria, which contain lipopolysaccharides, are more difficult to porate compared to membranes of Gram-positive bacteria, which are devoid of lipopolysaccharides (Piggot et al., 2011). These findings are consistent with our finding that gangliosides, which contain large sugar moieties, are more difficult to porate compared to other lipids in our membranes.

Atomistic MD studies of Gurtovenko and Lyulina showed that POPE bilayer is more difficult to porate than POPC (Gurtovenko and Lyulina, 2014). Lower poration propensity has been attributed to the primary amines in the POPE head groups capable of intra and intermolecular hydrogen bonding, in contrast to the choline moieties in the POPC head groups. POPE lipids are thus packed more densely than the POPC lipids, which hinders the penetration of water molecules in the bilayer and slows down the reorientation of the lipid head groups into the pore. Mixing these two lipids in an asymmetric bilayer (POPE in one and POPC in the other leaflet) results in an intermediate poration propensity with respect to pure POPC and POPE. Our average and brain plasma membranes contain an asymmetric mixture of PC and PE (as well as other) lipids; therefore, our analysis shows comparable poration propensity of PC and PE lipids.

Mixing different lipids does not always influence the poration propensity. Through atomistic MD simulations, Levine and Vernier showed that 30% of POPS lipids in one leaflet of an otherwise POPC bilayer does not have a significant effect on poration propensity compared to pure POPC (Levine and Vernier, 2012). Consistent with their findings, we did not see considerable influence of PS lipids on poration propensity.

Hu et al., determined the free energy for pore formation in 18 monocomponent coarse-grained Martini bilayers (Hu et al., 2015). They studied fully saturated and monounsaturated lipids with different chain lengths containing PC, PE, PG or PE headgroup. Their analysis confirmed that poration propensity depends on the architecture of both lipid headgroups and tails in coarse-grained membranes, similarly as in atomistic membranes. They found slightly lower poration propensity in monounsaturated lipids compared to fully saturated lipids. However, Breton and Mir showed that GUVs from polyunsaturated DLiPC lipids are considerably more prone to poration compared to monounsaturated DOPC lipids (Breton and Mir, 2018). The finding of Hu et al., and Breton and Mir are consistent with our finding that pores strongly favor polyunsaturated lipids compared to both fully saturated and monounsaturated lipids.

Interesting are also the measurements of van Uitert et al., on planar lipid bilayers, which shows that mixing polyunsaturated PE or π lipids with DPhPC lipids increases the poration propensity compared to pure DPhPC membranes (van Uitert et al., 2010). This shows how the architecture of lipid tails can become more important for poration than the architecture of lipid headgroups (as mentioned above, PE and π lipids are less prone for poration than PC lipids containing the same tail). These results support our finding that pores preferentially form in regions enriched with polyunsaturated lipids.

Influence of cholesterol and lipid order

*As* cholesterol is added to the system, the lipids organize in the liquid-ordered phase. The cholesterol organizes itself in the hydrophobic core of the bilayer, where it can condense the lipids and it can alter the mechanical properties of the membrane, such as the thickness, the bending stiffness and the fluidity. However, the addition of cholesterol does not always lead to the same results. Depending on the concentration of the cholesterol and the architecture of the lipid, cholesterol can either decrease or raise the electroporation threshold (Portet and Dimova, 2010). Also, mechanical studies on bilayers have shown the non-universal and lipid-specific effect of cholesterol (Gracià et al., 2010; Pan et al., 2008). Studies of Mauroy et al., have shown that an increasing concentration of cholesterol on POPC decreases the poration propensity, whereas this increased cholesterol shows no significant influence poration propensity of Egg PC (Mauroy et al., 2015). Similar results for Egg PC have been shown before by Portet and Dimova (Portet and Dimova, 2010). In addition, they reported that increasing cholesterol could increase poration propensity for DOPC vesicles. Surprisingly, the experimental results on the effect of cholesterol on *poration propensity* of different lipid bilayers have not been fully supported by MD simulations. Simulations on the effect of cholesterol on poration propensity of POPC show similar results as found experimentally on GUVs (Casciola et al., 2014). Nevertheless, MD simulations of Fernandez et al., on DOPC showed an decrease of *poration propensity* when adding cholesterol (Fernández et al., 2010), which is in disagreement with the experimental results on GUVs (Portet and Dimova, 2010). Overall, the influence of cholesterol on *poration propensity* of a lipid bilayer is non-universal and strongly dependent on the architecture of the lipids.

By mixing two different lipids together with cholesterol, coexisting liquid-ordered and liquid-disordered phases can occur in the membrane. Liquid-ordered domains contain saturated lipids and cholesterol and liquid-disordered domains contain unsaturated lipids and possibly a low level of cholesterol. Van Uitert et al., studied this effect of cholesterol on *poration propensity* in planar bilayers made from binary lipid mixtures (van Uitert et al., 2010). They observed that the effect of cholesterol on poration propensity depends on the cholesterol percentage. At low percentages, *poration propensity increased* slightly with respect to pure binary mixture without cholesterol. However, above a certain threshold percentage, *the poration propensity* decreased together with the increase in cholesterol. From the experimental results it is difficult to interpret the molecular mechanisms of this biphasic influence of cholesterol percentage on poration propensity. With MD simulations on heterogeneous membranes made from ternary mixtures, Reigada showed that the probability of pore formation is highest in the middle of the liquid disordered phase (Reigada, 2014). Reigada’s observations have been corroborated by Sengel and Wallace by visualizing pores in droplet-interface bilayers containing liquid-order and liquid-disordered domains (Sengel and Wallace, 2016). In agreement with the findings described above, we found that in the average and brain plasma membranes, pores tend to avoid regions enriched with cholesterol and prefer locations with lower lipid order.

Influence of membrane thickness and membrane area compressibility modulus

Atomistic MD simulations by Ziegler and Vernier indicated that for fluid-phase fully-saturated or monounsaturated lipids with a PC head group, *poration propensity tends to decrease* with membrane thickness and thus with increasing the chain length of the lipid tails (Ziegler and Vernier, 2008). Similarly, Hu et al., determined the free energy for pore formation in 18 monocomponent coarse-grained Martini bilayers (Hu et al., 2015) and showed correlation between the probability for poration and membrane thickness, with thinner membranes tending to be more easily porated (Author response image 1) . Note, however, that around a given value of thickness the spread of energy barriers for poration is considerable (Author response image 1) , showing that thickness is not the only parameter influencing poration, as the lipid headgroup and tail architecture have an influence as well. Consistent with the thickness not being the only factor governing poration, we see greater poration of average plasma membranes, which are slightly thicker than brain plasma membranes (~3.8 nm and ~3.5 nm, respectively). Similarly, we found that the local membrane thickness is not a good predictor of pore locations (note also that the variation of the local thickness in our membranes is small, only about ±0.25 nm). However, we found that for a given type of membrane, the poration propensity is greater for membranes that can be thinned more by the electric field. The change in membrane thickness caused by electric field is associated with an increase in the membrane area. The ability of a membrane to change its area under electric field is inversely proportional to the membrane area compressibility (or stretching) modulus. Needham and Hochmuth studied GUVs made from SOPC lipids, SOPC:cholesterol mixture, and lipids extracted from red blood cells. They showed that the electroporation propensity increases with decreasing membrane area compressibility modulus, consistent with our results (Needham and Hochmuth, 1989).

**Author response image 1. sa2fig1:** Correlation between membrane thickness and the free energy barrier for creating a 3-nm-radius pore in a lipid bilayer. Each dot corresponds to one of 18 monocomponent coarse-grained Martini membranes, made from fully saturated and monounsatured lipids with different chain lengths and different headgroups (PC, PE, PS, and PG). Reproduced from data tabulated in (Hu et al., 2015).

References

Breton M, Mir LM. 2018. Investigation of the chemical mechanisms involved in the electropulsation of membranes at the molecular level. *Bioelectrochemistry* 119:76–83. doi:10.1016/j.bioelechem.2017.09.005

Casciola M, Bonhenry D, Liberti M, Apollonio F, Tarek M. 2014. A molecular dynamic study of cholesterol rich lipid membranes: comparison of electroporation protocols. *Bioelectrochemistry*, Bio-Electroporation organised by COST TD1104 100:11–17. doi:10.1016/j.bioelechem.2014.03.009

Fernández ML, Marshall G, Sagués F, Reigada R. 2010. Structural and Kinetic Molecular Dynamics Study of Electroporation in Cholesterol-Containing Bilayers. *J Phys Chem B* 114:6855–6865. doi:10.1021/jp911605b

Gracià RS, Bezlyepkina N, Knorr RL, Lipowsky R, Dimova R. 2010. Effect of cholesterol on the rigidity of saturated and unsaturated membranes: fluctuation and electrodeformation analysis of giant vesicles. *Soft Matter* 6:1472–1482. doi:10.1039/B920629A

Gurtovenko AA, Lyulina AS. 2014. Electroporation of asymmetric phospholipid membranes. *J Phys Chem B* 118:9909–9918. doi:10.1021/jp5028355

Hu Y, Sinha SK, Patel S. 2015. Investigating hydrophilic pores in model lipid bilayers using molecular simulations: Correlating bilayer properties with pore-formation thermodynamics. *Langmuir* 31:6615–6631. doi:10.1021/la504049q

Levine ZA, Vernier PT. 2012. Calcium and Phosphatidylserine Inhibit Lipid Electropore Formation and Reduce Pore Lifetime. *J Membrane Biol* 245:599–610. doi:10.1007/s00232-012-9471-1

Mauroy C, Rico-Lattes I, Teissié J, Rols M-P. 2015. Electric Destabilization of Supramolecular Lipid Vesicles Subjected to Fast Electric Pulses. *Langmuir* 31:12215–12222. doi:10.1021/acs.langmuir.5b03090

Needham D, Hochmuth RM. 1989. Electro-mechanical permeabilization of lipid vesicles. Role of membrane tension and compressibility. *Biophys J* 55:1001–1009.

Pan J, Mills TT, Tristram-Nagle S, Nagle JF. 2008. Cholesterol Perturbs Lipid Bilayers Nonuniversally. *Phys Rev Lett* 100:198103. doi:10.1103/PhysRevLett.100.198103

Piggot TJ, Holdbrook DA, Khalid S. 2011. Electroporation of the *E. coli* and S. aureus membranes: Molecular dynamics simulations of complex bacterial membranes. *J Phys Chem B* 115:13381–13388. doi:10.1021/jp207013v

Polak A, Bonhenry D, Dehez F, Kramar P, Miklavčič D, Tarek M. 2013. On the Electroporation Thresholds of Lipid Bilayers: Molecular Dynamics Simulation Investigations. *J Membrane Biol* 246:843–850. doi:10.1007/s00232-013-9570-7

Polak A, Tarek M, Tomšič M, Valant J, Ulrih NP, Jamnik A, Kramar P, Miklavčič D. 2014. Electroporation of archaeal lipid membranes using MD simulations. *Bioelectrochemistry*, Bio-Electroporation organised by COST TD1104 100:18–26. doi:10.1016/j.bioelechem.2013.12.006

Portet T, Dimova R. 2010. A New Method for Measuring Edge Tensions and Stability of Lipid Bilayers: Effect of Membrane Composition. *Biophysical Journal* 99:3264–3273. doi:10.1016/j.bpj.2010.09.032

Reigada R. 2014. Electroporation of heterogeneous lipid membranes. *Biochimica et Biophysica Acta (BBA) – Biomembranes* 1838:814–821. doi:10.1016/j.bbamem.2013.10.008

Sengel JT, Wallace MI. 2016. Imaging the dynamics of individual electropores. *PNAS* 113:5281–5286. doi:10.1073/pnas.1517437113

van Uitert I, Le Gac S, van den Berg A. 2010. The influence of different membrane components on the electrical stability of bilayer lipid membranes. *Biochimica et Biophysica Acta (BBA) – Biomembranes* 1798:21–31. doi:10.1016/j.bbamem.2009.10.003

Vernier PT, Levine ZA, Gundersen MA. 2013. Water bridges in electropermeabilized phospholipid bilayers. *Proceedings of the IEEE* 101:494–504. doi:10.1109/JPROC.2012.2222011

Ziegler MJ, Vernier PT. 2008. Interface Water Dynamics and Porating Electric Fields for Phospholipid Bilayers. *J Phys Chem B* 112:13588–13596. doi:10.1021/jp8027726

The reviewers also raise a number of issues related to the validation of the coarse-grained model, presentation of technical details and wording in certain discussions.

We have addressed all Reviewers’ comments, as described on the following pages.

Reviewer #2 (Recommendations for the authors):I realize validation of the coarse-grained results are likely out of scope for this already quite extensive manuscript (i.e. currently a number of the lipids in the complex membranes used don't have AA force field parameters and a in-between complex mixture would have to be used, etc, etc), therefore, I tried to suggest the validation as future directions. I do have a number of more specific questions and comments listed below:One perception and/or wording comment. The brain is commonly thought to be enriched in polyunsaturated lipids. But you write that APM has more PU than the BPM:L100 "Compared with BPM, APM has lower fraction of cholesterol, higher fraction of phosphatidylcholine (PC) lipids and higher fraction of monounsaturated (MU) and polyunsaturated (PU) lipids in the top leaflet." andL391 "Compared with BPMs, APMs contain a smaller fraction of cholesterol and fully saturated lipids and a greater fraction of polyunsaturated lipids,"This looks to be correct from how you defined the PU group (including also double unsaturated tails), compared to the 0, 1, 2, 3+ grouping in Figure 1A in Ingolfsson et al., 2017 (ref 41). Also, same paper Table 1, shows that on average (for the lipid fraction) there are more unsaturations per tail in the BPM than in the APM. Maybe some explanation or redefinition could prevent confusion for future readers.

Indeed, we grouped lipids based on unsaturation in a different was as Ingolfsson et al., 2017. The reason for this is related to lipid oxidation and its role in electroporation, as we discuss in Section 3.2. We added the following explanation to Table 1:

* The way in which lipids are grouped by tail saturation is different than in (Ingólfsson et al., 2017), where the grouping is based on the total number of double bonds in both lipid tails. Here, the grouping is motivated by the role of lipid oxidation in electroporation, whereby lipids tails containing two or more double bonds are considerably more prone to oxidative damage than tails containing a single double bond. This is because bis-allylic hydrogens are much more easily abstracted by free radicals compared to allylic hydrogens (Reis and Spickett, 2012). Furthermore, membranes made of polyunsaturated lipids (by our definition) were found to be considerably more prone to poration/rupture by mechanical stretching compared to membranes made of lipids containing a single bond in one or both lipid tails (Olbrich et al., 2000). Thus, we consider that a lipid is polyunsaturated only if it contains at least one polyunsaturated tail.

From the first paragraph in Results we deleted the following sentence, which was in any case incomplete, because there are further differences in the compositions, visible in Figure 1 but not mentioned in the sentence:

“Compared with BPM, APM has lower fraction of cholesterol, higher fraction of phosphatidylcholine (PC) lipids and higher fraction of monounsaturated (MU) and polyunsaturated (PU) lipids in the outer leaflet.”

We also added the following sentence to Section 3.4:

“Note that the polyunsaturated lipids in BPMs have on average greater number of double bonds than in APMs (Ingólfsson et al., 2017); however, this is not enough to balance the poration rate with APMs.”

Somewhat ill-defined wording comment. When first reading the manuscript, I got the sense that the Bayesina analysis was giving different result than the lipid feature/ML analysis. After reading it more in-depth I understand that they show different results on different “things” propensity/time vs spatial location and therefore are complementary. Maybe some rewording in the manuscript can make this clearer. I think my biggest stumbling block was “however” instead of something like additionally.P1,L22, “learning. However, by” However ….P2 “However, by analysing poration kinetics with Bayesian survival analysis we then show that”

As suggested, we changed the word “however” to “additionally”.

I really like the analysis in section 2.4, but confused by one thing. It looks like form Figure 5 – sup. 1 and Figure 5B that the model is unable to capture the initial delay in pore formation time, the initial ~2.5 ns discussed later due to the bilayer expansion. Is this my misunderstanding or due to granularity of the binned data and/or lack of x-d offset term in β?

We thank the reviewer for the positive feedback. The delay time the reviewer is referring to is possibly the short initial time interval where there are no poration events (roughly the first nanosecond). This time is not captured by our model since it is smaller than our discretization grid. In contrast, ~3 ns discussed in the manuscript is the time the system takes to reach the steady state kinetics, the time at which *l*_0_(*t*) plateaus. During the pre-steady state regime poration does occur but at a lower rate.

Please note that we found an error in the script for plotting *l*_0_(*t*), which has now been corrected.

P16, L354 “Since membranes are practically volume-incompressible, the increase in area is directly related to a decrease in thickness.”. Did you check this in your case? i.e. this is true for actual bilaye area, and projected area for very small flat bilayers but not larger undulating bilayers. I know both the APM and BPM were equilibrated with Z restrains to make them mostly flat but Ingolfsson et al., 2017 Figure 4A shows they are not totally flat and the BPM has more local undulations.

We verified that the increase in the projected membrane area is correlated with a decrease in membrane thickness, whereby we determined the average membrane thickness by averaging the thickness values computed at all lipid positions by MemSurfer. We also estimated the change in membrane volume, by summing the products of area per lipid *a* and half of the membrane thickness *d*, computed at each lipid position by MemSurferV≈∑iNlipidsaidi2

where *N_lipids_* is the number of all lipids in the membrane. The relative changes in projected area, average thickness, and volume for different magnitudes of nonporating electric fields are shown in Figure 5 —figure supplement 4. The relative change in membrane volume is indeed nonzero; however, it is much lower than the relative change in projected area or membrane thickness.

Figure 5 —figure supplement 5 shows the steady state values of the relative change in membrane thickness and how they depend on the electric field *E*, similarly as shown for projected membrane area in Figure 5 —figure supplement 6. The relative change in membrane thickness decreases linearly with *E*^2^.

Finally, we verified that the factors *b_i_* are inversely correlated with the relative change in membrane thickness, as shown in Figure 5D. Therefore, we can confirm that a greater poration rate is associated with membrane thinning.

Based on the comment, we made the following modifications to the manuscript:

– We added Figure 5 —figure supplement 4.

– We added Figure 5 —figure supplement 5.

– We added Figure 5D.

– We made slight modifications to the text to accomodate the additional results.

– We changed the problematic sentence in Section 3.3 from "Since membranes are practically volume-incompressible, the increase in area is directly related to a decrease in thickness.” to ”The increase in membrane area is directly related to a decrease in membrane thickness (Figure 5 —figure supplement 4).”

P16, L355 "Membrane thinning is expected to facilitate pore formation, as water molecules need to travel shorter distance when bridging the membrane.". Yes, but is it possible to utilize the power of the simulations and check for water bridging more directly? Here I realize this might be outside the scope of this manuscript, but have you considered looking at local bilayer properties such as "defects" (water penetration to the tails) or local min water distance across the bilayer (water dependent and not lipids dependent like the very related bilayer thickness)?

This is an excellent suggestion. We have thought about looking at water defects, but we first wanted to focus on the relationship between bilayer properties in the absence of an electric field and the probability of pore formation when exposing the membrane to an electric field. This is what we would in general like to do – predict the most likely poration sites in a membrane with a new composition without additional extensive simulations and analyses. As we obtained reliable prediction of porated and nonporated locations with machine learning models by just looking at local lipid densities, we decided to leave it at that for the present study.

We expect that the rate of formation of water defects depends on the electric field strength, similarly as the rate of formation of transmembrane water bridges and pores. Therefore, monitoring water defects in the absence of an electric field might turn out uninformative. Another variable that is presumably related to occurrence of water defects and worth further investigation is the local lateral pressure in the headgroup and tail region, which has already been correlated with poration rate in simpler atomistic bilayer systems (Polak et al., 2013, http://dx.doi.org/10.1016/j.bioelechem.2013.12.006; Gurtovenko and Lyulina, 2014, http://dx.doi.org/10.1021/jp5028355). We think this will be very interesting to study in the future, especially in combination with simulations on corresponding atomistic systems obtained after backmapping regions of interests from larger coarse-grained systems.

P9, L215+. I was quite confused reading that paragraph (L215 to L226), granted I am not an ML expert, but maybe some further explanation is in order. Also, could the questions of overfitting vs need for more data for higher accuracy be solved by increasing the number of depolarizable/hyperpolarizable simulation repeats? and/or leaving out some data for validation checks?

We have addressed comments R2.6 and R2.12 together. Please refer to R2.12 for our response.

Reviewer #3 (Recommendations for the authors):1. One aspect that remains unclear in this work relates to how universal the reported findings are and whether the large fraction of species present in the simulations is essential to the reported outcome, especially regarding the dependence on membrane elasticity. Indeed, to establish this correlation, it would have been interesting, using the selected approaches to first explore simpler membranes, and in particular, directly compare the poration probability and kinetics as a function of composition and membrane leaflet asymmetry. Membrane compositions with only a few representatives of the major classes of components would have been a good start as this would allow direct comparison to experimental results and would be helpful to resolve the role of the main actors involved.

We thank the reviewer for the comment, and we fully agree that it will be very interesting to compare coarse-grained simulations on simpler systems with experiments on membranes with corresponding lipid composition, especially with the aim of carefully validating coarse-grained simulations for assessing electroporation.

However, probing what governs electroporation in simpler systems has already been performed, leading us to focus our efforts on complex systems. The literature review presented in R0.1 shows that there are many different factors influencing poration in simpler membranes and it its quite impossible to predict which ones will be most important in a complex mixture. Consequently, our study provides a way to fill this gap. The literature review, and our comments along, further show that our findings are fully in agreement with previous findings on simpler system.

To clarify this, we added the following text to Section 3.1:

“Overall, studies on simple model systems showed that there are many different factors influencing poration and it is quite impossible to predict which ones will be most important in a complex mixture such as the plasma membrane.

Our findings from membranes composed of >60 different lipid types mimicking the realistic composition of plasma membranes are fully in agreement with findings from simpler model systems.”

Regarding the dependence of poration probability on membrane elasticity: Our results show that the electroporation propensity increases with decreasing membrane area compressibility (or stretching) modulus. This is in agreement with experimental results of Needham and Hochmuth 1989 (https://dx.doi.org/10.1016%2FS0006-3495(89)82898-X), who studied electroporation thresholds in GUVs made from SOPC lipids, SOPC:cholesterol mixture, and lipids extracted from red blood cells.

In the revised manuscript, we have added the reference to the paper of Needham and Hochmuth in Section 3.3:

“Inverse correlation between electroporation propensity and membrane area compressibility modulus has also been found experimentally (Needham and Hochmuth, 1989).”

To motivate further studies combining simulations and experiments, in accordance with suggestions made by Reviewer #2, we added the following sentence at the end of Section 3.6.

“Nevertheless, we hope our study will motivate further exploration on the validity of coarse-grained membrane models for studying membrane electroporation, both in comparison with corresponding atomistic computational membrane models and experimental model membrane systems.”

2. The introduction (paragraph starting at line 70) explains that experimentally, pores in lipid membranes are difficult to assess. Work on giant unilamellar vesicles showing direct imaging of pores in lipid membranes suggests the opposite and should be referenced here (see e.g. DOI: 10.1529/biophysj.104.050310 and DOI: 10.1002/advs.202004068).

We agree. The following sentence, together with suggested references, has been added:

Pores have been imaged in giant unilamellar vesicles (Lira et al., 2021; Riske and Dimova, 2005); however, these pores have reached sizes on the order of 1 mm, which have not been observed in cell plasma membranes, likely because the actin cytoskeleton limits pore expansion (Perrier et al., 2019).

3. It will be good to explicitly discuss the following: The poration of the membranes in the simulations occurs within the first 15 nanoseconds, implying that the findings apply to pulses of high amplitude and nanosecond duration, but not necessarily to pulses of lower field strength and in the micro- or millisecond range duration (i.e. including conditions used in medical applications). A comment about this is due.

As a response to comment (R1.1) made by Reviewer 1 we now explain the relationship between the electric field used in simulations and experiments in Section 5.1, among which we also address comment R3.3:

“While the conditions in our study strictly correspond to nanoseconds-long exposures to electric field, we expect similar behaviour also for longer electric pulses. Nevertheless, on a longer time scale, electrodeformation of the cell membrane and/or increases in local membrane curvature caused by electric field might play additional roles in the pore formation process (Dimova et al., 2009; Perrier et al., 2017).”

4. The authors indicate that such short poration times are needed to minimize lateral diffusion and allow mapping of local membrane features. Presumably, lipid diffusion and mixing which is substantial on longer times scales, might jeopardize the validity of the findings for longer pulse duration. Could the authors introduce a discussion also about this aspect?

The membranes that we used in our study have already been equilibrated by Ingolfsson et al., in an 80 μs-long simulation, therefore the lipid mixing is at equilibrium when we start the simulation. Of course, if we kept the simulation of a chosen membrane, e.g. APM #1, running for tens of microseconds before applying an electric field, the local lipid organization would be somewhat different since domains laterally move along the membrane. However, our results from 4 APM membranes, all containing slightly different lipid organizations, are consistent, indicating that our results are still valid even if we consider some additional waiting time before pore creation starts.

The situation with the waiting time mentioned above resembles the situation when a cell is exposed to an electric field. From an electrical point of view, a cell behaves like a capacitor. Therefore, the transmembrane voltage gradually increases with time until it becomes sufficiently high to trigger pore creation. Author response image 2 shows the time course of electroporation at the pole of a cell (*θ* = 0), predicted by a model developed by DeBruin and Krassowska in 1999, doi: 10.1016/S0006-3495(99)76973-0. As soon as the transmembrane voltage exceeds certain value (~1.3 V in the model), pores are rapidly created, i.e. the pore density exhibits a jump. This is because pore creation rate depends exponentially on the transmembrane voltage (or the local electric field strength within the membrane). With increasing number of pores, the membrane conductivity also increases, which in turns starts decreasing the transmembrane voltage and stops pore creation. This model therefore shows that most pores are created on a nanosecond time scale, no matter how long the pulse is.

**Author response image 2. sa2fig2:** Time course of the induced transmembrane voltage, pore density, and membrane conductivity at the cell pole (*q* = 0) upon exposure to 1-ms-long electric pulse. Only the first 10 ms after the onset of the electric pulse are shown for clarity. The results are reproduced from the paper by DeBruin and Krassowska 1999, doi: 10.1016/S0006-3495(99)76973-0.

Consequently, we have no reason to think that lipid diffusion and mixing on longer times scales would jeopardize the validity of the findings for longer pulse duration.

5. The authors should explicitly specify the solution in which the membranes are simulated and the ionic strength.

Thank you for noticing that we forgot to report the NaCl concentration. We added a sentence to Section 5.1:

“The NaCl concentration was ~150 mM.”

6. The finding that gangliosides are quite important for poration (Figure 4B) is very interesting. The authors should discuss possible reasons and potential implications. Along these lines (even though not directly comparable), there has been a recent report on the poration of GM1-doped vesicles which exhibit much longer pore lifetimes compared to PC membranes (DOI: 10.1073/pnas.1722320115).

Our study shows that pores avoid domains enriched with gangliosides. This is expected according to previous atomistic simulations on phospholipid and archaeal lipid membranes by Polak et al., 2014 (Ref. 54), which showed that large sugar moieties in the lipid headgroups make pore formation more difficult, which we already discussed in Section 3.1. We unfortunately cannot comment on pore lifetimes, as we have not investigated the kinetics of pore closure, nor have we investigated bilayers without gangliosides.

7. The authors should explicitly specify whether the size of the simulation box adjusts to accommodate the area of the pores in the membrane and whether more than one pore are simultaneously detected in one membrane patch. In line 379, the authors state that one can assume that *A*_*i*_ ≈ *A*_*j*_ because all membranes have the same total area. Do they refer to the area before or after the application of the field? Could the authors further clarify the connection between the number of possible pore nucleation sites (as suggested on line 373) and the total area of the membranes?

The following sentence about adjustments of the box size was added to Section 5.1:

“The pressure coupling allows the size of the simulation box to adjusts to the changes in the membrane area.”

The information about multiple simultaneous poration events was already in Section 5.2:

“In up to ~20% simulations, two or more pores formed practically simultaneously. In such a case, we considered all of these pores for analysis.”

When assuming *A_i_* ≈ *A_j_* we consider the area before poration. Nevertheless, due to the exponential dependence of the pore formation rate on the energy barrier for pore formation, small variations in the membrane area are not very influential.

The theoretical description of the pore creation rate is based on ideas from the classical nucleation theory, where a pore is considered as the nucleus of a new phase. In oversimplified terms, each lipid is considered as a possible nucleation site. We refer to the following references, which provide further useful information:

Glaser, R W, S L Leikin, L V Chernomordik, V F Pastushenko, and A I Sokirko. ‘Reversible Electrical Breakdown of Lipid Bilayers: Formation and Evolution of Pores’. *Biochimica et Biophysica Acta* 940, no. 2 (24 May 1988): 275–87. https://doi.org/10.1016/0005-2736(88)90202-7.

Evans, Evan, and Volkmar Heinrich. ‘Dynamic Strength of Fluid Membranes’. *Comptes Rendus Physique* 4, no. 2 (March 2003): 265–74. https://doi.org/10.1016/S1631-0705(03)00044-6. (See their Section 3)

Vasilkoski, Zlatko, Axel T. Esser, T. R. Gowrishankar, and James C. Weaver. ‘Membrane Electroporation: The Absolute Rate Equation and Nanosecond Time Scale Pore Creation’. *Physical Review E* 74, no. 2 (3 August 2006): 021904. https://doi.org/10.1103/PhysRevE.74.021904.

8. Similarly, how does the pore size compare to the size of the data points displayed in Figure 2. In the caption of Figure 2, to avoid confusion, the authors should specify that all points correspond to the first poration event.

We did not evaluate the pore size, as all pores expand upon creation until the system explodes, which is typical for simulations at constant electric field strength and is related to the relatively small system size. The size of points in Figure 2 was also coding for poration time, similarly as the color of the points. To avoid confusion, we corrected the figure and made all points equal size.

We also added the following sentence to Figure 2 caption:

“All data points correspond to the first poration event.”

9. Figure 3 —figure supplement 3: The authors state that these quantities are computed from equation (1), however the caption above the graphs indicates that these two plots correspond to data where either no field was applied or a non-porating field was applied (i.e., situations where no pore is supposed to form in the membrane). How were these two diagrams plotted as they compare non porated with porated locations? Please clarify this in the caption.

Thank you for the comment. We had mistakenly referred to equation (2) instead of equation (7), which probably led to the confusion. This has now been corrected.

The figure caption already gives information on how the distances were obtained:

Distances between probability density estimates of individual features in porated and nonporated locations, calculated according to equation (7). … The graphs on the left side correspond to values extracted from a 10-ns-long trajectory before electric field application. The graphs on the right side correspond to values extracted from a 10-ns-long trajectory, where a non-porating electric field of +106.3 mV/nm (analysis for depolarization) or -106.3 mV/nm (analysis for hyperpolarization).

10. The data for change in the membrane area as a function of the square of the nonporating field strength (Figure 5D and supplement 4) should be discussed in terms of the stretching elasticity modulus of the membranes. Typical stretching elasticity moduli of single-lipid membranes lie around 250 mN/m (see e.g. DOI: 10.1016/S0006-3495(00)76295-3). To claim the validity of the reported correlation between poration and mechanics, it has to be clarified, whether the applied simulation force fields can be used to correctly reflect the membrane elasticity. To address this (if not already reported for the selected force fields), the authors should measure the stretching elasticity for single-lipid membranes and compare with experimental values.

In the manuscript we referred to the membrane area compressibility modulus in the last paragraph of Section 2.4. We now also added the term “stretching” modulus:

We found that the values of *β*_*i*_ correlate with the ability of a membrane to thin and expand its area under electric field, i.e., they corelate inversely with the membrane area compressibility (or stretching) modulus (Figure 5D,E).

The membrane area compressibility modulus *K_A_* has been evaluated for coarse-grained Martini bilayers previously. For DPPC bilayer in fluid phase with size comparable to our membranes, Marrink et al., 2004 (https://doi.org/10.1021/jp036508g) obtained *K_A_* = 260 ± 40 mN/m, which compares well with the experimental estimate for DPPC *K_A_* = 250 of Nagle et al., 2000 (https://doi.org/10.1016/S0304-4157(00)00016-2). For POPC, Chacón et al., 2015 (http://dx.doi.org/10.1063/1.4926938) obtained a value of *K_A_* = 310 ± 20 mN/m, which is close to 272 mN/m reported by Janosi and Gorfe 2010 (https://doi.org/10.1021/ct100381g) for atomistic POPC bilayers described with Charmm force-field, and K = 277 ± 10 mN/m reported by Braun et al., 2013 (https://doi.org/10.1021/jp401718k) for a similar DOPC lipid described by united atom force field. It is also close to the experimental value of 265 ± 18 mN/m measured in DOPC giant unilamellar vesicles by Rawicz et al., 2000 (https://doi.org/10.1016/S0006-3495(00)76295-3).

11. Line 270: "is fairly time independent" should be reformulated and justified by further discussion. As the authors refer to the values of β_i of Figure 5 – Supplement figure 2, the trend in this figure shows two main problems: (i) the values are positive for the whole time interval which is inconsistent with Figure 5 showing only negative values, and (ii) the values look actually time dependent as for instance APM-hyp shows a drop of 25% of its initial value on the time interval where its related probability density displayed Figure 5 A is non zero. Please discuss this issue.

We thank the Reviewer for the valuable comments.

(i) During the revision we found a bug in the plotting code used to make Figure 5 —figure supplement 2. The corrected plot is shown in Author response image 1. After correcting the code, the *β*_*i*_(*t*) values of the time-dependent model are closer to those of the time-independent model. Nevertheless, the values of *β*_*i*_ in the time-independent and time-dependent model cannot be directly compared.

We have added the following explanation to Figure 5 —figure supplement 2 caption:

“Note that the value of *β*_*i*_(*t*) are different from the ones obtained in the time-independent model in Figure 5. Since all coefficients are parametrized simultaneously with all the data, different combinations of magnitudes of *λ*_0_(*t*) and *β*_*i*_ can result in an equivalent *λ*_*i*_(*t*). Therefore, in order to compare how rates increase or decrease in two different systems i and j, only the relative change can be studied *λ*_*i*_(*t*)/*λ*_*j*_(*t*) = exp(*β*_*i*_)/exp(*β*_*j*_).”

For completeness, we also added the plot of *λ*_0_(*t*) for the time-dependent *β* model.

(ii) We agree with the reviewer that this should have been explained in more detail. We have included the following explanations to Section 2.4:

“The *β*_*i*_(*t*) obtained were found to slightly drop in the interval 0 < *t* < 3 ns until becoming practically time-independent (Figure 5 —figure supplement 2). Given that the *β*_*i*_(*t*) for all systems follow the same trend and that the time-dependent model only adds information in the initial stage of electroporation and has higher uncertainty, the time-independent model was selected for its easier interpretability and lower complexity.”